# The yeast endocytic early/sorting compartment exists as an independent sub-compartment within the *trans*-Golgi network

**Junko Y Toshima[1,2]\*, Ayana Tsukahara[2], Makoto Nagano[2], Takuro Tojima[3], Daria E Siekhaus[4†], Akihiko Nakano[3], Jiro Toshima[2]\***

[1]School of Health Science, Tokyo University of Technology, Tokyo, Japan; [2]Department of Biological Science and Technology, Tokyo University of Science, Tokyo, Japan; [3]Live Cell Super-Resolution Imaging Research Team, RIKEN Center for Advanced Photonics, Saitama, Japan; [4]Institute of Science and Technology Austria, Klosterneuburg, Austria

**\*For correspondence:**
toshimajk@stf.teu.ac.jp (JYT);
jtosiscb@rs.tus.ac.jp (JT)

**Present address:** †Department of Molecular, Cell and Developmental Biology, University of California, Los Angeles, Los Angeles, CA 90095, United States

**Competing interest:** The authors declare that no competing interests exist.

**Abstract** Although budding yeast has been extensively used as a model organism for studying organelle functions and intracellular vesicle trafficking, whether it possesses an independent endocytic early/sorting compartment that sorts endocytic cargos to the endo-lysosomal pathway or the recycling pathway has long been unclear. The structure and properties of the endocytic early/sorting compartment differ significantly between organisms; in plant cells, the *trans*-Golgi network (TGN) serves this role, whereas in mammalian cells a separate intracellular structure performs this function. The yeast syntaxin homolog Tlg2p, widely localizing to the TGN and endosomal compartments, is presumed to act as a Q-SNARE for endocytic vesicles, but which compartment is the direct target for endocytic vesicles remained unanswered. Here we demonstrate by high-speed and high-resolution 4D imaging of fluorescently labeled endocytic cargos that the Tlg2p-residing compartment within the TGN functions as the early/sorting compartment. After arriving here, endocytic cargos are recycled to the plasma membrane or transported to the yeast Rab5-residing endosomal compartment through the pathway requiring the clathrin adaptors GGAs. Interestingly, Gga2p predominantly localizes at the Tlg2p-residing compartment, and the deletion of GGAs has little effect on another TGN region where Sec7p is present but suppresses dynamics of the Tlg2-residing early/sorting compartment, indicating that the Tlg2p- and Sec7p-residing regions are discrete entities in the mutant. Thus, the Tlg2p-residing region seems to serve as an early/sorting compartment and function independently of the Sec7p-residing region within the TGN.

## Editor's evaluation

In this study, the authors use high-speed and high-resolution imaging to investigate the role of the yeast syntaxin homolog Tlg2p in endocytic vesicle sorting. They obtain compelling data to show that the Tlg2p-residing compartment within the trans-Golgi network functions as an early/sorting compartment, where endocytic cargos are sorted to either the recycling pathway or the endo-lysosomal pathway. The authors also describe additional molecular details of this sorting process and overall provide important insights into the mechanism of endocytic vesicle sorting in budding yeast.

## Introduction

Clathrin-mediated endocytosis is the best-characterized type of endocytosis in eukaryotic cells and plays crucial roles in many physiological processes (*Kaksonen and Roux, 2018*; *Mettlen et al., 2018*). After leaving from the plasma membrane (PM), a clathrin-coated vesicle (CCV) is uncoated and transported to the early/sorting compartment (*Cullen and Steinberg, 2018*; *Paez Valencia et al., 2016*). In yeast, the molecular mechanisms regulating this CCV formation and internalization have been well characterized, but it still remains unclear how and where uncoated endocytic vesicles are delivered to the early/sorting compartment. A recent study has reported that budding yeast lacks distinct early endosomes and that the TGN where Sec7p resides is the first destination for endocytic traffic, functioning as an early endosome-like compartment (*Day et al., 2018*). However, several other reports have indicated that yeast has two distinct types of endosomal compartments, one containing yeast Rab5 (Vps21p) and the other containing yeast Rab7 (*Lachmann et al., 2012*). Additionally, we have previously demonstrated that endocytosed cargos are only slightly co-localized with Sec7p-containing TGN cisternae (*Toshima et al., 2014*), classified into the early-to-late TGN (*Tojima et al., 2019*). These inconsistent observations are likely attributable to technical difficulties in visualizing the cargo-sorting process and have complicated our understanding of the properties of the yeast early/sorting compartment.

The yeast R-SNAREs (soluble N-ethylmaleimide-sensitive factor attachment protein receptors), Snc1p and Snc2p, are yeast orthologs of vesicle-associated membrane protein (VAMP) and were originally identified as proteins required for the fusion of secretory vesicles with the PM via Q-SNAREs Sso1p and Sso2p (*Gerst et al., 1992*; *Protopopov et al., 1993*). Snc1p contains a conserved endocytosis signal, which is recognized by the clathrin adaptor protein and is thereby endocytosed with the CCV (*Grote et al., 2000*; *Gurunathan et al., 2000*). Disruption of this endocytic signal causes defects in the internalization of Snc1p itself and other endocytic cargos, such as the fluorescent endocytic tracer FM4-64 and the α-factor receptor Ste2p (*Gurunathan et al., 2000*). This result suggests that Snc1p and Snc2p function as R-SNAREs in endocytic pathways by interacting with Q-SNAREs. Tlg1p and Tlg2p have high homology to syntaxins and bind Snc1p and Snc2p (*Abeliovich et al., 1998*; *Holthuis et al., 1998b*; *Paumet et al., 2001*). Since these Q-SNAREs are localized to the TGN and putative early endosomes, these proteins have been considered to play a role in transport between the TGN and endosomal compartments (*Holthuis et al., 1998a*; *Lewis et al., 2000*). Tlg1p and Tlg2p are also known to co-localize with endocytosed FM4-64 soon after internalization (*Abeliovich et al., 1998*; *Dobzinski et al., 2015*). From these observations, it has been suggested that Tlg1p and/or Tlg2p are Q-SNAREs-mediating fusion between endocytic vesicles and early endosomal compartments.

Recent studies using high-speed, high-resolution, and four-dimensional (4D) time-lapse imaging have revealed that in yeast intra-Golgi cargo trafficking from the early Golgi to the TGN is mediated by cisternal maturation (*Kurokawa et al., 2019*; *Losev et al., 2006*; *Matsuura-Tokita et al., 2006*). Detailed localization analyses of various Golgi/TGN-resident proteins have shown that the Golgi-TGN transition gradually proceeds with sequential recruitment of these proteins (*Tojima et al., 2019*). In this process, Tlg2p appears at the TGN earlier than Sec7p, a marker of the early-to-late TGN, and also disappears before Sec7p from the TGN (*Tojima et al., 2019*). This observation suggests that endocytic vesicles containing Snc1p R-SNARE might target the TGN compartment where Tlg2p resides, although this has not yet been proven directly.

In our previous study, we demonstrated that fluorescently labeled endocytic cargo, yeast mating pheromone α-factor (Alexa-α-factor), accumulates at clathrin-coated pits and is internalized with the CCV (*Toshima et al., 2006*). Within 5 min after Alexa-α-factor internalization, Alexa-α-factor-labeled endosomal compartments become co-localized with the yeast Rab5, Vps21p, and fuse with each other, resulting in the formation of enlarged endosomal compartments (*Toshima et al., 2014*). However, it has been unclear whether Alexa-α-factor is directly transported to the Vps21p-residing compartment from the endocytic vesicle. We recently showed that although endocytosis is not essential, post-Golgi vesicle transport is crucial for Vps21p-mediated endosome formation (*Nagano et al., 2019*). Thus, as the TGN seems to play a key role in endocytic cargo transport, it is important to clarify how the TGN regulates endocytic cargo transport to the Vps21p-residing compartment. Here we succeeded in simultaneous triple-color and 4D (3D plus time) observation to visualize endocytic cargo together with the Tlg2p-residing region by super-resolution confocal live imaging microscopy (SCLIM) (*Kurokawa*

*and Nakano, 2020*; *Tojima et al., 2023*). We show that Alexa-α-factor endocytosed by the CCV is incorporated directly into the Tlg2p-residing sub-compartment within the TGN, and then moves on from there when another TGN-representative protein Sec7p appears. Such visualization, along with genetic manipulations of endocytic pathway components, suggests that the Tlg2p-residing sub-compartment within the TGN is the primary endocytic-accepting region that serves as an early/sorting compartment that sorts endocytosed cargo to further destinations.

## Results

### Endocytosed α-factor is transported to the Tlg2p-residing sub-compartment within the TGN

Sec7p, one of the yeast guanine-nucleotide exchange factors (GEFs) for Arf GTPases, is known as a representative TGN marker (*Casler et al., 2022*; *Kurokawa et al., 2019*). A previous study has reported that the Sec7p-residing TGN compartment is the first destination for endocytic traffic and functions as an early endosome-like sorting compartment (*Day et al., 2018*). However, since Tlg2p, which is a putative Q-SNARE for the endocytic vesicle (*Abeliovich et al., 1998*; *Paumet et al., 2001*; *Séron et al., 1998*), exhibits temporal localization patterns distinct from Sec7p (*Tojima et al., 2019*), the exact timing and locus of endocytic vesicle targeting remained ambiguous. As a first step toward clarifying this, we compared the localization of Alexa594-labeled α-factor (A594-α-factor) with GFP-tagged Tlg2p and Sec7p, whose functionalities have been previously confirmed (*Llinares et al., 2015*; *Séron et al., 1998*). The yeast Golgi and TGN are highly dynamic organelles whose cisternae rapidly change their composition with an approximate maturation rate of less than 1 min (*Matsuura-Tokita et al., 2006*; *Tojima et al., 2019*). Therefore, we defined 'overlapping' as a distance of less than 129 nm (2 pixels) between two peaks of GFP and mCherry/Alexa594 intensity in 2D imaging. At 5 min after internalization, A594-α-factor began to accumulate at several intracellular puncta, although the majority still remained on the PM (*Figure 1A*). GFP-fused Tlg2p was detected in several nonuniform structures, such as puncta or tubules, with different sizes and shapes (*Figure 1A and E*), and ~71% of them clearly overlapped with A594-α-factor-labeled puncta at 5 min after A594-α-factor internalization (*Figure 1B*). A faint fluorescent signal of GFP-Tlg2p was also observed in the cytosol, and we confirmed that this signal was located in the vacuolar lumen by labeling the vacuolar membrane with FM4-64 (*Figure 1—figure supplement 1A*). This vacuolar localization was not visible in some cells, possibly due to the location of the vacuole. After 10 min, A594-α-factor-labeled puncta had increased in both number and fluorescent intensity, and ~77.7% of GFP-Tlg2p overlapped with them (*Figure 1A and B*). In contrast, Sec7-GFP, which was also detected as several punctate structures, showed considerably lower overlap with A594-α-factor signals than GFP-Tlg2p either at 5 min (26.0 ± 9.4%) or 10 min (22.7 ± 6.0%) after A594-α-factor internalization (*Figure 1B*, *Figure 1—figure supplement 1B*). After 20 min, the number of A594-α-factor-labeled puncta decreased as the α-factor was transported to the vacuole via the pre-vacuolar compartments (PVCs) (*Toshima et al., 2014*; *Figure 1A*). At this time point, most of the A594-α-factor was localized inside the vacuole and at puncta around the vacuole, and substantial GFP-Tlg2p was also localized at these puncta labeled by A594-α-factor (47.7 ± 3.9%), whereas Sec7-GFP was rarely localized there (5.0 ± 5.4%) (*Figure 1A and B*, *Figure 1—figure supplement 1B*). These observations suggest the localization of GFP-Tlg2p at the early-to-late stage endosomes, as well as the TGN. We also compared the localization of A594-α-factor with Tlg1p, another putative Q-SNARE for the endocytic vesicle, that is reported to localize mainly at the early endosomes and partially at the TGN (*Holthuis et al., 1998b*). Similar to GFP-Tlg2p, GFP-Tlg1p highly overlapped with A594-α-factor-labeled puncta (74.4 ± 5.9%) at 5 min after A594-α-factor internalization (*Figure 1B*, *Figure 1—figure supplement 1C*). However, after 10 min the fraction of A594-α-factor overlapping with GFP-Tlg1p decreased to ~51.1% and further decreased to ~26.1% after 20 min (*Figure 1B*, *Figure 1—figure supplement 1C*). GFP-Tlg1p signals overlapped well with mCherry-Tlg2p-labeled puncta, except at puncta around the vacuole (*Figure 1—figure supplement 1D*), suggesting that Tlg1p localizes at the TGN and early-stage endosomes, as described previously (*Holthuis et al., 1998b*). To confirm that α-factor overlaps with both Tlg1p and Tlg2p at the TGN just after internalization, we conducted triple-color imaging with GFP-Tlg1p, mCherry-Tlg2p, and Alexa647-labeled α-factor (A647-α-factor). As expected, the majority of endocytosed A647-α-factor localized at the compartments where both Tlg1p and Tlg2p are present at 5 min after internalization

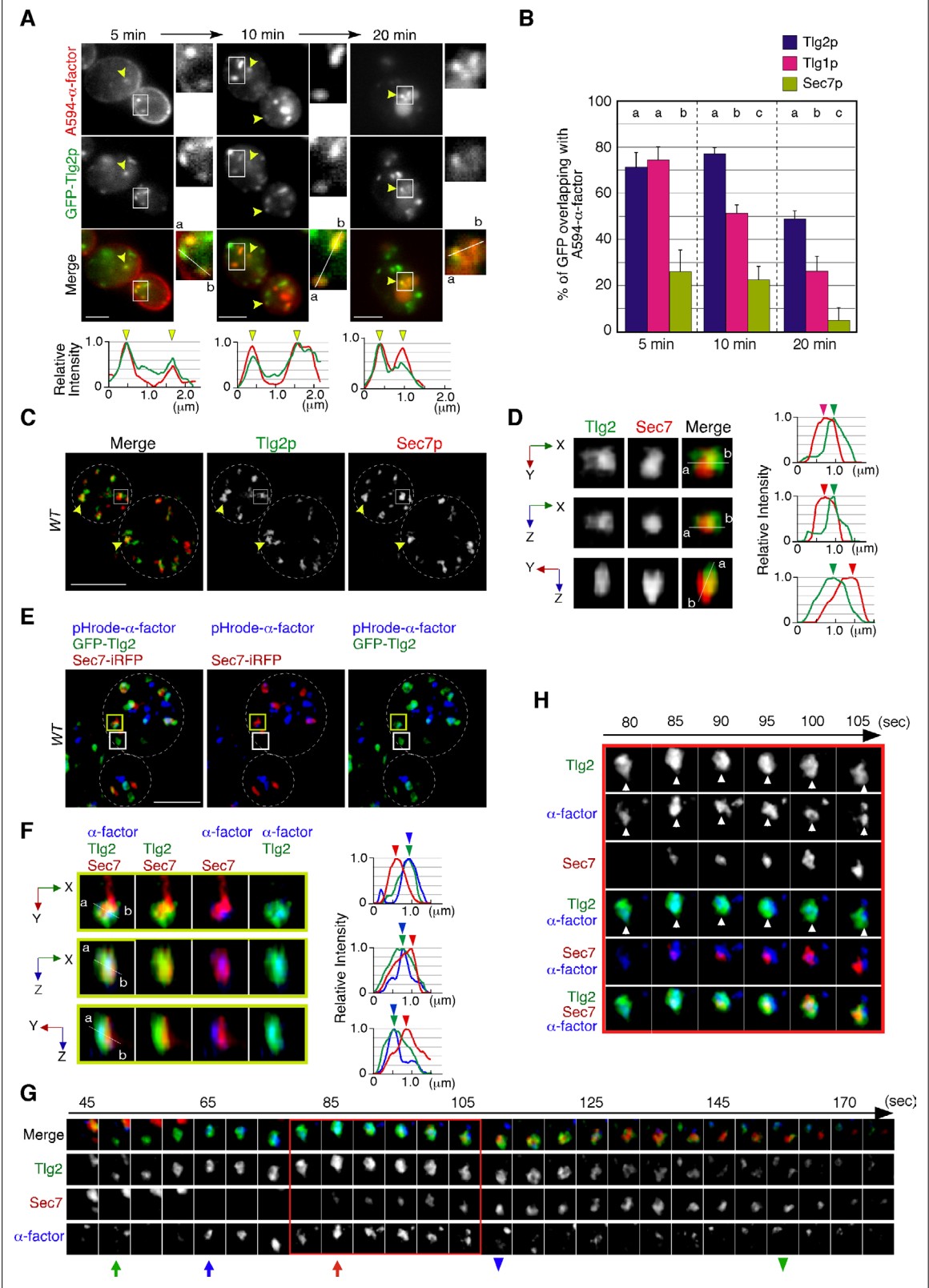

**Figure 1.** Localization of endocytosed α-factor at the Tlg2p-residing compartment. (**A**) 2D imaging of A594-α-factor and GFP-Tlg2p. Arrowheads indicate examples of overlapping localization. Representative fluorescence intensity profiles along a line (direction from 'a' to 'b') are indicated in the lower panels. (**B**) Quantification of GFP-Tlg2p, GFP-Tlg1p, and Sec7-GFP overlapping with A594-α-factor. Data show the mean ± SEM from n ≥ 3 experiments (n > 30 puncta for each experiment). Different letters indicate significant differences at p<0.05 between the indicated times (i.e., no

*Figure 1 continued on next page*

*Figure 1 continued*

significant difference for a vs. a, significant difference for a vs. b with p<0.05), one-way ANOVA with Tukey's post hoc test. Error bars indicate the standard SD from n ≥ 3 experiments (n ≥ 30 puncta for each experiment). (**C**) 3D super-resolution confocal live imaging microscopy (SCLIM) imaging of GFP-Tlg2p and Sec7-mCherry. White dashed lines indicate cell edges. (**D**) Multi-angle magnified 3D views of the boxed area and the representative fluorescence intensity profiles. Line scan as in (**A**) shown at right. (**E**) 3D SCLIM imaging of GFP-Tlg2p, Sec7-iRFP, and pHrode-α-factor; boxed areas shown magnified in (**F–H**). The images were acquired simultaneously at 5 min after pHrode-α-factor internalization. (**F**) Multi-angle magnified 3D views of the yellow-boxed area in (**E**). Line scan as in (**A**) shown at right. (**G**) Time series of region in the white-boxed area in (**E**). Arrows and arrowheads denote the appearance and disappearance of each marker. (**H**) Higher-magnification views of the red-boxed area in (**G**). Scale bars, 2.5 µm.

The online version of this article includes the following source data and figure supplement(s) for figure 1:

**Source data 1.** Data for graphs presented in *Figure 1B*.

**Figure supplement 1.** Localization of α-factor, Sec7p, Tlg1p, and Tlg2p in wild-type cells.

(*Figure 1—figure supplement 1E*). These observations suggest that α-factor is transported to a region distinct from the Sec7p-residing TGN compartment where Tlg1p and Tlg2p localize, and then from there moves to the PVC via the endocytic pathway.

A recent study has demonstrated that the timing of Tlg2p recruitment to the TGN is earlier than that of Sec7p (*Tojima et al., 2019*). To examine whether Tlg2p localizes at a separate region from the Sec7p-residing region within the TGN, we performed simultaneous dual-color 3D analysis of these proteins using SCLIM. As shown in *Figure 1C*, we observed two types of GFP-Tlg2p localization, one at regions in which only GFP-Tlg2p was visible, and the other at locations adjacent to the Sec7p-residing region. A faint fluorescent signal in the vacuolar lumen (*Figure 1—figure supplement 1A*) was not observed by SCLIM, presumably because the fluorescence signal was too weak. Previous studies have reported that Tlg2p cycles between late Golgi and endosomal compartments, and thus localizes to both compartments (*Lewis et al., 2000*). It has also been shown that at the TGN Tlg2p appears earlier than Sec7p and also disappears before Sec7p from the TGN (*Tojima et al., 2019*). Therefore, we considered the structures where only GFP-Tlg2p localized to be either an endosomal compartment or the TGN compartment before Sec7-mCherry came in. Localization of GFP-Tlg2p and Sec7-mCherry at the TGN was analyzed by line scan using the xy, xz, or yz planes, which revealed that GFP-Tlg2p signal partially overlaps with the TGN compartment labeled by Sec7-mCherry, but is mostly spatially separated (*Figure 1D, Video 1*). This observation suggests that the Tlg2p-residing structure exists as a sub-compartment that is distinct from the Sec7p-residing structure within the TGN (hereafter referred to as Tlg2p or Sec7p sub-compartment, respectively). We next performed simultaneous triple-color 3D imaging to determine whether α-factor localized to the Tlg2p or the Sec7p sub-compartment. For triple-color imaging with GFP-Tlg2p and Sec7-iRFP, we labeled α-factor with pHrode Red (pHrode-α-factor), whose fluorescent wavelength can be separated from those of GFP and iRFP. We also confirmed that expression of Sec7-iRFP has no effect on cell growth (*Figure 1—figure supplement 1F*). pHrode-α-factor clearly labeled endocytic compartments after internalization to the same extent as A594-α-factor (*Figure 1E*). 3D SCLIM imaging, similar to the 2D analysis, revealed that at 5 min after internalization pHrode-α-factor signals overlapped with the GFP-Tlg2p signals, around half of which were adjacent to Sec7-iRFP-labeled regions (*Figure 1E*). Line scan analyses revealed that the peak of pHrode-α-factor signal coincides with that of GFP-Tlg2p signal rather than Sec7p (*Figure 1F, Video 1*). To further understand the spatiotemporal relationship between the Tlg2p and Sec7p sub-compartments during α-factor transport, we

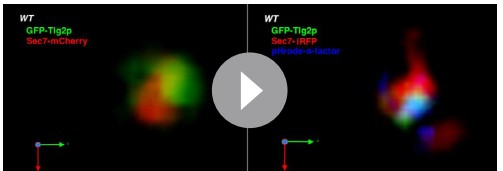

**Video 1.** Left: multi-angle 3D reconstructed movie of GFP-Tlg2p (green) and Sec7-mCherry (red) in wild-type cell. Right: multi-angle 3D reconstructed movie of GFP-Tlg2p (green), Sec7-iRFP (red), and pHrode-α-factor (blue).

https://elifesciences.org/articles/84850/figures#video1

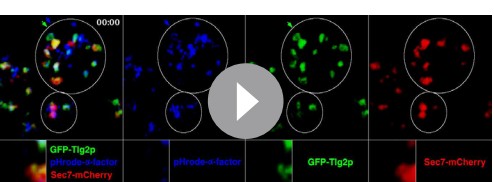

**Video 2.** Triple-color 4D movie of GFP-Tlg2p (green), Sec7-iRFP (red), and pHrode-α-factor (blue) in a wild-type cell. Arrows indicate examples of the sequential appearance and disappearance of each protein.

https://elifesciences.org/articles/84850/figures#video2

performed 4D (3D plus time) imaging by SCLIM. Consistent with the previous study (*Tojima et al., 2019*), we observed that Sec7-iRFP appeared in the vicinity of GFP-Tlg2p, and then the signal of GFP-Tlg2p gradually disappeared, while that of Sec7-iRFP increased (*Figure 1G*, *Video 2*). During this process, pHrodo-α-factor became joined to the preexisting Tlg2p sub-compartment and stayed there for several tens of seconds, and then disappeared soon after Sec7-iRFP arrival (*Figure 1G*, *Video 2*). Magnified images from the time series shown in *Figure 1G* indicate that pHrodo-α-factor already co-localizes with GFP-Tlg2p at the time when Sec7p shows up (*Figure 1H*). This observation suggests that the Tlg2p sub-compartment adjacent to the Sec7p sub-compartment plays a specific role in the endocytic pathway.

## Endocytic vesicles interact directly with the Tlg2p-residing sub-compartment

Co-localization of α-factor with the Tlg2p sub-compartment at the early stage of endocytosis motivated us to examine whether the Tlg2p sub-compartment is a direct target of endocytic vesicles. We had shown previously that endocytic vesicles labeled with the endocytic vesicle markers Sla1- or Abp1-GFP moved to the A594-α-factor-labeled compartment, which is considered to be the early endosome (*Toshima et al., 2006*). We had also demonstrated that Alexa 488-labeled α-factor accumulates at the Abp1p patch on the cell surface and is internalized concomitantly (*Toshima et al., 2006*). To examine whether endocytic vesicles move to the Tlg2p sub-compartment, we tagged Abp1p with mCherry and observed the dynamics of Abp1-mCherry and GFP-Tlg2p. The functionality of mCherry-tagged Abp1p was confirmed by testing its ability to grow in a *sla1Δ* background (*Figure 2—figure supplement 1A*) as a combination of *abp1Δ* and *sla1Δ* mutations results in synthetic lethality (*Holtzman et al., 1993*). Our current simultaneous dual-color 2D imaging showed that when Abp1-mCherry-labeled vesicles began moving, many of them traveled to Tlg2p sub-compartments and disappeared after arrival there (*Figure 2A*, *Video 3*). However, some Abp1-mCherry signals disappeared before reaching the Tlg2p sub-compartments, presumably because F-actin was disassembled from endocytic vesicles before reaching there. We then utilized the *arp3-D11A* mutant, which shows a delay in endocytic vesicle formation and a severe defect in vesicle internalization (*Martin et al., 2005*), to examine whether we could observe any Tlg2p sub-compartments approaching the endocytic site when endocytic vesicle movement is blocked. As described previously, in the *arp3-D11A* mutant Abp1p's lifetime was remarkably extended, indicating that the formation and internalization of endocytic vesicles were also severely impaired (*Figure 2B*, *Video 3*). Similar to the previous findings obtained when A594-α-factor was used as an early endosome marker (*Toshima et al., 2006*), the Tlg2p sub-compartment was observed to approach and make contact with the Abp1-mCherry-labeled vesicle remaining on the PM, and then disappeared (*Figure 2B*, *Video 3*).

To further examine the contact between endocytic vesicles and the Tlg2p sub-compartment, we performed dual-color 4D observation by SCLIM. As shown in *Figure 2C and D*, we found that several Abp1-mCherry patches attach to the Tlg2p sub-compartment in both wild-type cells and those from the *arp3-D11A* mutant. Time-lapse imaging of a wild-type cell revealed that an Abp1-mCherry-labeled endocytic vesicle appeared, stayed for 3–6s around the Tlg2p sub-compartment, and then disappeared (*Figure 2E*, *Video 4*). In contrast, in the *arp3-D11A* mutant, an Abp1-mCherry-labeled vesicle stayed around the Tlg2p sub-compartment more than 10s and then disappeared (*Figure 2F*, *Video 5*). Because the temporal resolution of SCLIM is 3s in this observation, we were unable to track all the Abp1-mCherry signals until their disappearance. The timing of dissociation of Abp1p from the endocytic vesicles depends on the speed of actin depolymerization around the vesicle (*Toret et al., 2008*) and thus is not constant: some Abp1p signals disappear after reaching the Tlg2p sub-compartment, but others disappear before. Despite this, we observed similar dynamics in at least 18% of Abp1-mCherry-labeled vesicles internalized in wild-type cells (*Figure 2H*). We also utilized Vps21p as a marker of another endosomal compartment. In contrast to the Tlg2p sub-compartment, GFP-Vps21p-labeled endosomal compartments often fused with each other, forming larger structures, and Abp1-mCherry-labeled vesicles rarely coalesced into them (*Figure 2G and H*). To further test if the Tlg2p sub-compartment interacting with Abp1-mCherry-labeled vesicles was within the TGN, we performed triple-color 4D imaging, including Sec7-iRFP. As Abp1-mCherry labels F-actin associated with endocytic vesicles (*Kaksonen et al., 2003*), Abp1-mCherry signals were not constant and varied in size and intensity according to the degree of actin polymerization, some Abp1-mCherry patches

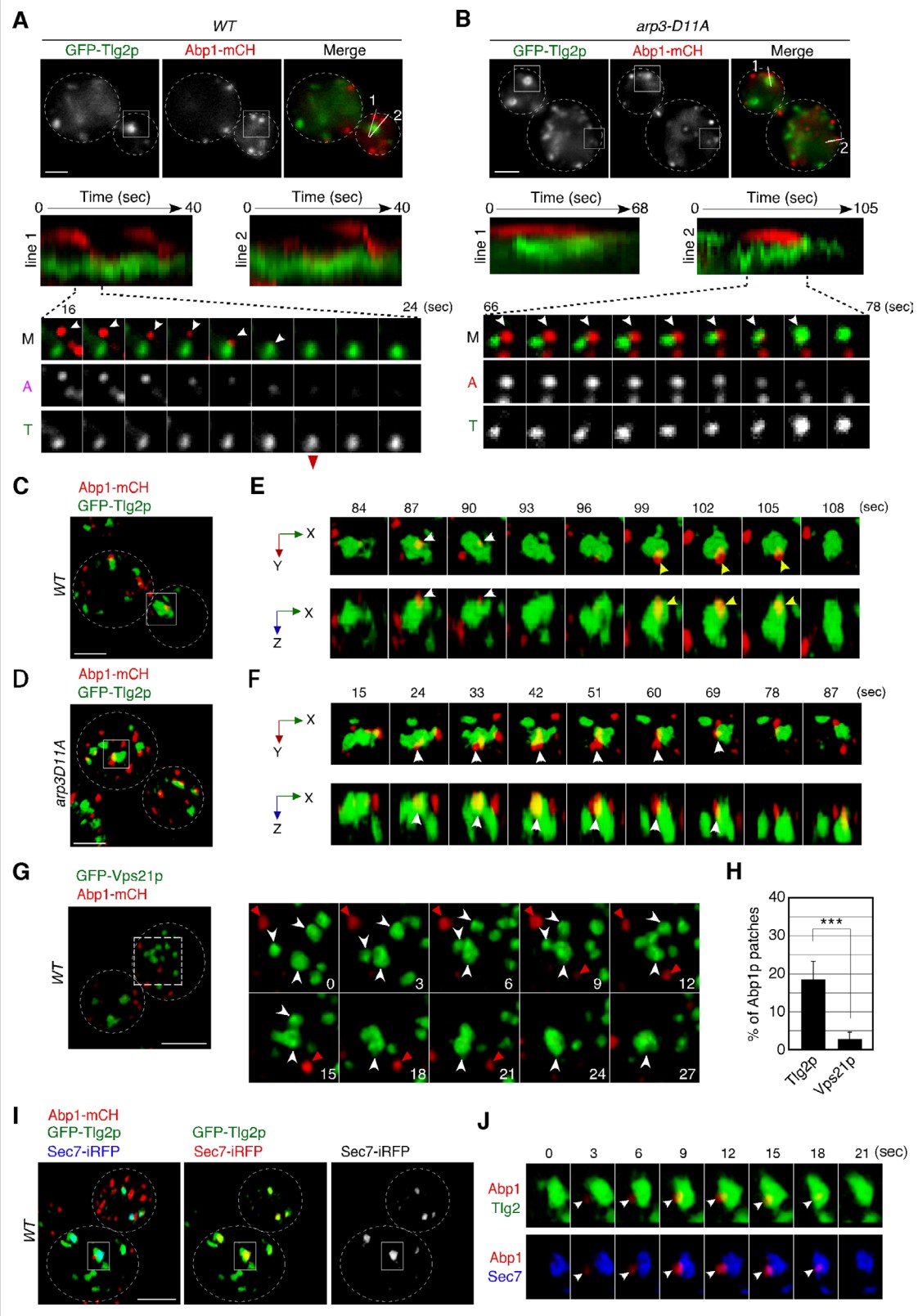

**Figure 2.** Dynamic behavior of the Tlg2p-residing compartment, endocytic vesicles, and cargos. (**A, B**) 2D imaging of GFP-Tlg2p and Abp1-mCherry in wild-type (**A**) and *arp3-D11A* cells (**B**).Kymographs along lines in the upper merged image are shown in the panels below. A time series of the boxed area in (**A**) and (**B**) is shown under the kymograph. Arrowheads highlight the movement of Abp1p toward the Tlg2p sub-compartment. (**C–F**) 4D super-resolution confocal live imaging microscopy (SCLIM) imaging of GFP-Tlg2p and Abp1-mCherry in wild-type (**C, E**) and *arp3-D11A* (**D, F**). (**E, F**) Time

*Figure 2 continued on next page*

*Figure 2 continued*

series of the areas boxed in (**C**) and (**D**) are shown in multi-angle magnified 3D views. White and yellow arrowheads indicate the dynamics of different Abp1p patches. (**G**) 4D SCLIM imaging of GFP-Vps21p and Abp1-mCherry in wild-type cells. Time series of the boxed areas are shown in the right panels. White and red arrowheads indicate the dynamics of the Vps21p-residing endosome and Abp1p patch. (**H**) The percentages of Abp1p patches that disappeared from the Tlg2p- or Vps21p-residing compartment. Error bars indicate the SD from n ≥ 10 biological replicates (n ≥ 40 Abp1p patches for each experiment). ***p<0.001, unpaired *t*-test with Welch's correction. (**I**) 4D SCLIM imaging of wild-type cells expressing GFP-Tlg2p, Sec7-iRFP, and Abp1-mCherry. (**J**) Time series of the boxed area in (**I**) are shown in magnified 3D views. Arrowheads indicate the incorporation of Abp1p patches to the sub-compartment including GFP-Tlg2p and Sec7-iRFP signals. Scale bars, 2.5µm.

The online version of this article includes the following source data and figure supplement(s) for figure 2:

**Source data 1.** Data for graphs presented in *Figure 2H*.

**Figure supplement 1.** Functionality of Abp1-mCherry and dynamics of Abp1-mCherry patch.

fusing with each other (*Figure 2—figure supplement 1B–D*). However, we found that some Abp1-mCherry-labeled vesicles stayed and disappeared on the Tlg2p sub-compartment adjacent to the Sec7p sub-compartment (*Figure 2I and J*). These results support the idea that the initial destination of endocytic vesicles is the Tlg2p sub-compartment.

## α-Factor transported to the Tlg2p sub-compartment moves to the Vps21p-residing endosome, together with Tlg2p

We previously demonstrated that Vps21p localizes predominantly at endosomal compartments (*Toshima et al., 2014*). Co-localization between A594-α-factor and GFP-Tlg2p at 5–10min after internalization (*Figure 1B*) prompted us to investigate whether Vps21p also localizes to the Tlg2p sub-compartment. To this end, we imaged GFP-Tlg2p and mCherry-Vps21p, whose functionality has been previously confirmed (*Toshima et al., 2014*), simultaneously by 2D epi-fluorescence microscopy, and found that~53% of GFP-Tlg2p signals overlaps with mCherry-Vps21p (*Figure 3A and B*). This overlapping localization was observed at the Vps21p-residing smaller structures (*Figure 3A*, white arrowheads) and the larger Tlg2p sub-compartments (*Figure 3A*, yellow arrowheads). We also found that~26% of the Sec7-mCherry-labeled sub-compartment overlapped with GFP-Vps21p signals, suggesting that Vps21p partially localizes around the TGN as well as in endosomal compartments. Interestingly, we observed that mCherry-Vps21p-labeled vesicles associate with and move around the Tlg2p sub-compartment over 10s, and line scan analyses revealed that the Vps21p signal well overlaps with the GFP-Tlg2p signal (*Figure 3C*). Triple-color 3D imaging demonstrated that mCherry-Vps21p signals often made contact with the Tlg2p sub-compartment adjacent to the Sec7p sub-compartment (*Figure 3D and E*, *Video 6*). This interaction seemed to be transient because mCherry-Vps21p attached to

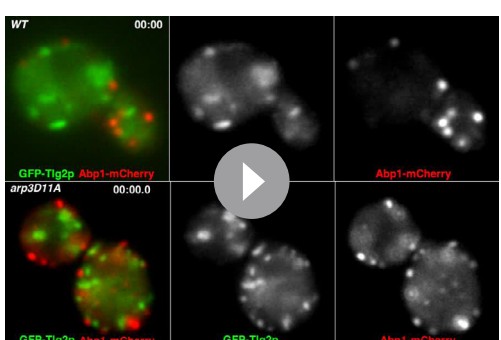

**Video 3.** Upper video: 2D time-lapse movie of Abp1-mCherry (red in merge) and GFP-Tlg2p (green in merge) in wild-type cell. Arrows indicate movement of an Abp1-mCherry-labeled endocytic vesicle toward the GFP-Tlg2p-labeled sub-compartment. Lower video: 2D time-lapse movie of Abp1-mCherry (red in merge) and GFP-Tlg2p (green in merge) in the *arp3-D11A* mutant. Arrows indicate movement of the GFP-Tlg2p-labeled sub-compartment toward an Abp1-mCherry-labeled endocytic vesicle.

https://elifesciences.org/articles/84850/figures#video3

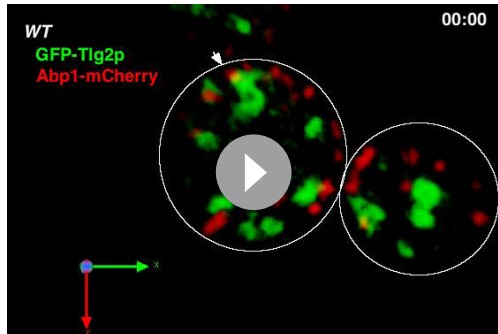

**Video 4.** Dual-color 4D movie of GFP-Tlg2p (green) and Abp1-mCherry (red) in a wild-type cell. Arrows indicate examples of Abp1-mCherry-labeled vesicles disappearing on the GFP-Tlg2p-labeled sub-compartment.

https://elifesciences.org/articles/84850/figures#video4

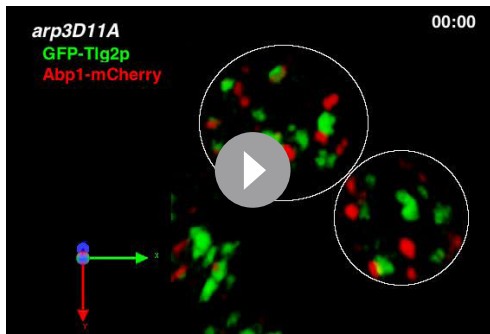

**Video 5.** Dual-color 4D movie of GFP-Tlg2p (green) and Abp1-mCherry (red) in the *arp3-D11A* mutant. Arrows indicate examples of Abp1-mCherry-labeled vesicles disappearing onto the GFP-Tlg2p-residing sub-compartment.

https://elifesciences.org/articles/84850/figures#video5

the Tlg2p sub-compartment for several tens of seconds and then became detached from there (*Figure 3F*, *Video 6*). Additionally, we observed that small GFP-Tlg2p signals moved together with mCherry-Vps21p when the mCherry-Vps21p-labeled puncta detached from the Tlg2p sub-compartment (*Figure 3G*, *Video 6*). Thus, the Vps21p-residing structures appear to transiently contact the Tlg2p sub-compartment, transporting Tlg2p to the Vps21p-residing compartment.

We next conducted 2D imaging using GFP-Tlg2p, mCherry-Vps21p, and A647-α-factor to determine the order of α-factor delivery. At 5–20min after internalization, the rate of A647-α-factor overlapping with GFP-Tlg2p and/or mCherry-Vps21p changed in a time-dependent manner (*Figure 3H and I*). We conducted quantitative analysis, categorizing A647-α-factor localization as overlapping with Tlg2p only (α-factor and Tlg2p), with Vps21p only (α-factor and Vps21p), with both of them (α-factor and Tlg2p & Vps21p), or as α-factor alone. This revealed that the overlap of α-factor signals with GFP-Tlg2p was the highest at 5min and then gradually decreased from 10 to 20min, whereas that with mCherry-Vps21p increased from 5 to 20min (*Figure 3H and I*). This result suggested that α-factor transported to the Tlg2p sub-compartment moves to the Vps21p-residing compartment in the endocytic pathway. At 15min after internalization, α-factor mostly reached the PVC (*Toshima et al., 2014*), and at the same time a portion of the GFP-Tlg2p signals was still co-localized with A647-α-factor and mCherry-Vps21p (*Figure 3H*), suggesting that Tlg2p exits the TGN and is then transported to the PVC via the Vps21p-residing compartment. To confirm this, we examined whether Tlg2p accumulates in endosomal intermediates observed in the mutant lacking the yeast Rab5 paralogs, *VPS21* and *YPT52* (*Toshima et al., 2014*). As shown previously, A594-α-factor accumulated in multiple endosomal intermediates in the *vps21Δ ypt52Δ* mutant, and Tlg2p was well localized there (*Figure 3J and K*). In the *vps21Δ ypt52Δ* mutant, we also observed that A594-α-factor accumulated at endosomal intermediate-like structures that did not contain the GFP-Tlg2p signal (*Figure 3J*, white arrowhead). It is noteworthy that the number of Tlg2p puncta increased (*Figure 3J*) while that of Sec7p puncta did not change in the mutant (*Toshima et al., 2014*). These observations support the idea that Tlg2p is transported to the PVC via the Vps21p-residing compartment.

## GGA adaptors are required for transport of endocytic cargo from the Tlg2p sub-compartment to the Vps21p-residing compartment

A previous study has reported that after reaching the TGN, α-factor is transported to the PVC, dependent on the TGN-resident Golgi-associated, γ-adaptin ear containing, Arf-binding protein (GGA) adaptors, Gga1p and Gga2p (*Day et al., 2018*). GGA adaptor was also shown to be clathrin adaptor and mediate TGN-to endosome traffic (*Black and Pelham, 2000*). Therefore, we speculated that deletion of *GGA1* and *GGA2* might cause accumulation of α-factor at the TGN compartment where α-factor is located. In agreement with previous observations, endocytosed A594-α-factor in the *gga1Δ gga2Δ* mutant accumulated at several puncta in the cytoplasm at 20min after internalization, whereas in wild-type cells it was mostly transported to the vacuole at this time (*Figure 4A*). AP-1 complex is another clathrin adaptor, and epsin-related Ent3p/Ent5p act as accessories for GGAs and AP-1 adaptors (*Copic et al., 2007*; *Duncan et al., 2003*). Accumulation of A594-α-factor was also observed in cells lacking Ent3p and Ent5p (*Nagano et al., 2019*), but not observed in cells lacking Apl4p, a subunit of the AP-1 complex (*Figure 4A*), indicating that the AP-1-dependent pathway is not essential for sorting of α-factor to the vacuole. By comparing the localization of A594-α-factor with the Sec7-GFP-labeled sub-compartment, we found that the regions where A594-α-factor accumulated differed between the mutants (*Figure 4B*). In the *gga1Δ gga2Δ* mutant, localization of Sec7-GFP overlapping with A594-α-factor at this resolution (<130nm) was markedly increased to 55.7 ± 5.8%, whereas it

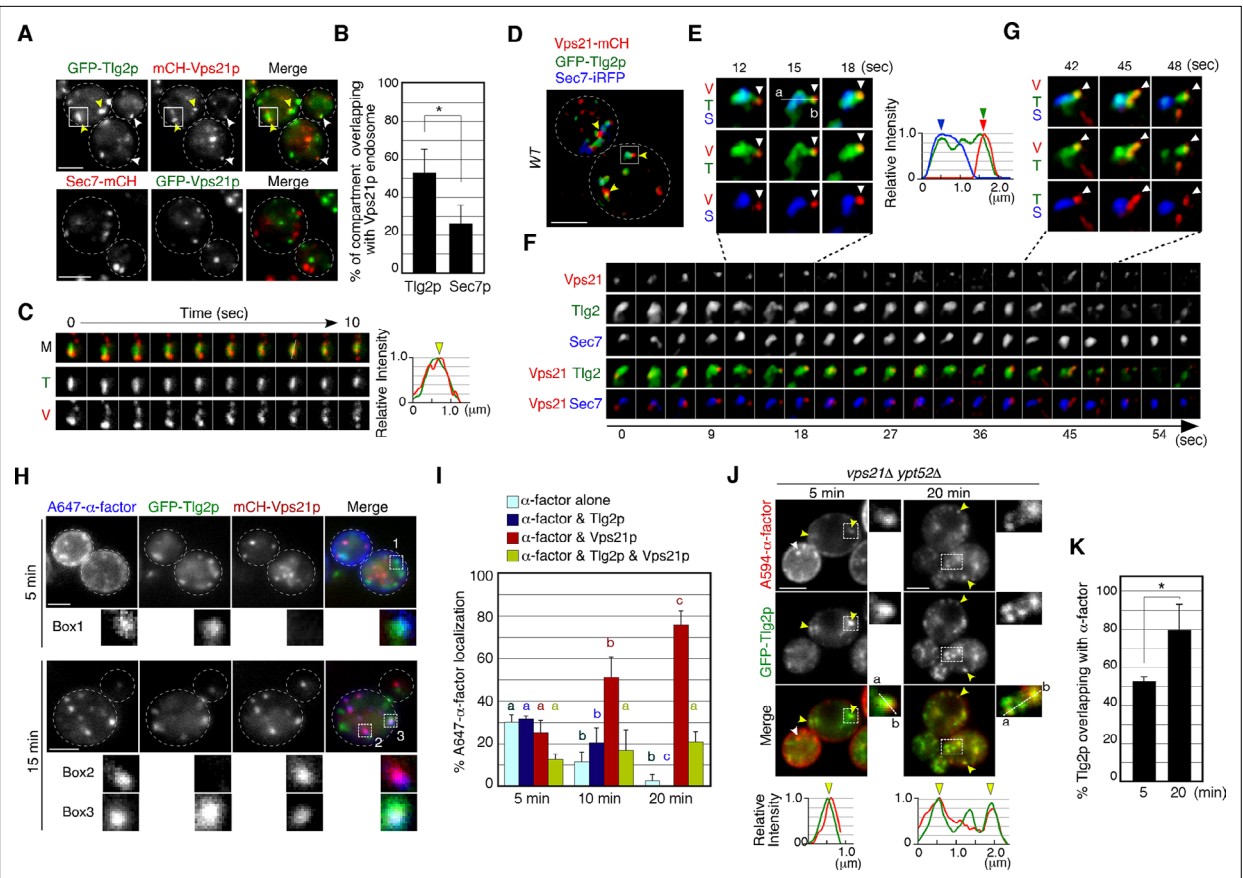

**Figure 3.** Alexa-α-factor is transported from the Tlg2p-residing compartment to the Vps21p-residing endosomal compartment. (**A**) 2D imaging of GFP-Tlg2p or Sec7-mCherry and mCherry/GFP-Vps21p. (**B**) Quantification of Tlg2p or Sec7p overlapping with Vps21p. Error bars indicate the SD from n ≥ 3 experiments (n≥30 puncta for each experiment). (**C**) Time series of the region in the boxed area in (**A**). Representative fluorescence intensity profiles along a line in the merged image at 8s are shown on the right. Yellow arrowhead indicates overlapping localization. (**D**) 4D super-resolution confocal live imaging microscopy (SCLIM) imaging of GFP-Tlg2p, Sec7-iRFP, and mCherry-Vps21p. Arrowheads indicate examples of the association of GFP-Tlg2p and mCherry-Vps21p. (**E**) Magnified views from the time-series in (**F**). Arrowheads indicate a Vps21p-residing endosome. Representative fluorescence intensity profiles along a line in the merged images at 15s are shown to the right. (**F**) Time series of the region in the boxed area in (**D**). (**G**) Further magnified views from the time series in (**F**). (**H**) 2D imaging of A647-α-factor, GFP-Tlg2p, and mCherry-Vps21p. The images were acquired at 5 and 15min after A647-α-factor internalization. Higher-magnification views of the boxed areas are shown in the lower panels. (**I**) Quantification of A647-α-factor overlapping with GFP-Tlg2p or mCherry-Vps21p. Data show the mean ± SEM from n ≥ 3 experiments (n > 30 puncta for each experiment). Comparisons are made between the same colors, with different letters indicating significant difference (p<0.05) between the indicated times, one-way ANOVA with Tukey's post hoc test. (**J**) 2D imaging of A594-α-factor and GFP-Tlg2p in *vps21Δ ypt52Δ* cells. The images were acquired at 5 and 20min after A594-α-factor internalization. Higher-magnification views of the boxed area are shown in the right panels. Arrowheads indicate examples of the overlapping localization of A594-α-factor and GFP-Tlg2p. (**K**) Quantification of GFP-Tlg2p overlapping with A594-α-factor. Error bars indicate the SD from n ≥ 3 experiments (n > 30 puncta for each experiment). *p<0.05, unpaired *t*-test with Welch's correction. Scale bars, 2.5µm.

The online version of this article includes the following source data for figure 3:

**Source data 1.** Data for graphs presented in *Figure 3B*.

**Source data 2.** Data for graphs presented in *Figure 3I*.

**Source data 3.** Data for graphs presented in *Figure 3K*.

was only slightly increased in the *ent3Δ ent5Δ* mutant (11.7 ± 4.1%), relative to that in wild-type cells (2.6 ± 2.3%) at 20min after A594-α-factor internalization (*Figure 4B and C*). These results indicate a requirement for GGA adaptors in the export of A594-α-factor from the regions near the Sec7-GFP-labeled TGN compartment.

Since A594-α-factor is first transported to the Tlg2p sub-compartment, we speculated that A594-α-factor might accumulate at the region distinct from the Sec7p sub-compartment in the *gga1Δ gga2Δ* mutant. Thus, we examined the temporal changes in A594-α-factor localization in the *gga1Δ*

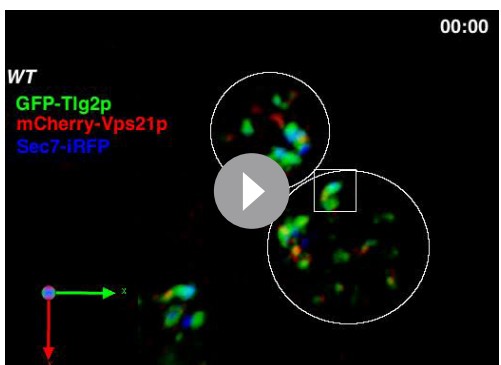

**Video 6.** Triple-color 4D movie of GFP-Tlg2p (green), mCherry-Vps21p (red), and Sec7-iRFP (blue) in a wild-type cell. Arrows indicate examples of association between GFP-Tlg2p and mCherry-Vps21p.
https://elifesciences.org/articles/84850/figures#video6

*gga2Δ* mutant by comparing the localization with GFP-Tlg2p. In contrast to wild-type cells, in the *gga1Δ gga2Δ* mutant, the overlap between A594-α-factor and GFP-Tlg2p increased over time (*Figure 4D and E*). We further classified the distance between the two peaks into those<65nm (1 pixel) and those from 65 to 130nm (1–2 pixels), and found that more than half of the α-factor accumulated at sites neighboring the GFP-Tlg2p peak (1–2 pixel) (33.5 ± 11.8%, 34.5 ± 9.0%, 57.0 ± 15.3%, and 51.1 ± 10.2% at 5, 10, 20, and 40min, respectively) (*Figure 4E*). To examine α-factor localization with higher spatial resolution, we observed the localization of pHrode-α-factor and GFP-Tlg2p at the Golgi/TGN by dual-color 3D SCLIM. Similar to images observed by 2D epi-fluorescence microscopy, we found that α-factor accumulates at the Tlg2p-residing region and its adjacent region (*Figure 4F–H*). Furthermore, triple-color 3D imaging demonstrated that pHrode-α-factor signal mostly co-localized with GFP-Tlg2 rather than Sec7-iRFP (*Figure 4I and J*, *Video 7*). These results suggest that in the *gga1Δ gga2Δ* mutant α-factor accumulates in or around Tlg2p sub-compartments due to the defective transport out of the compartment.

## GGA adaptors are required for turnover of the Tlg2p sub-compartment

GGA adaptors have been believed to function at the Sec7p-residing TGN compartment (*Daboussi et al., 2012*; *Dell'Angelica et al., 2000*) but our finding that α-factor accumulates at the Tlg2p sub-compartment in the *gga1Δ gga2Δ* mutant suggests that GGA may be required for the transport from the Tlg2p sub-compartment instead of the Sec7p sub-compartment. Thus, we next investigated the effects of deleting *GGA1* and *GGA2* on the Tlg2p or Sec7p sub-compartment. As the distribution of the Sec7p-residing TGN is known to vary during the cell division cycle (*Preuss et al., 1992*), we first examined the localization of Tlg2p and Sec7p at different cell cycle stages. We found that the number of Sec7p-residing puncta were slightly more abundant in M-phase cells than in S-phase cells (*Figure 5—figure supplement 1A and B*), but the rate of overlap between GFP-Tlg2p and Sec7-mCherry in the two phases was almost the same (*Figure 5—figure supplement 1C and D*). Simultaneous dual-color 2D imaging revealed that in the *gga1Δ gga2Δ* mutant the Tlg2p and Sec7p sub-compartments were more segregated and that the overlap of GFP-Tlg2p with Sec7-mCherry was markedly decreased (46.3 ± 2.6%), relative to wild-type cells (76.9 ± 7.7%) (*Figure 5A and B*). GFP-Tlg2p was also slightly localized at the vacuolar membrane, suggesting that retrograde transport of Tlg2p from the endosome to the TGN is partially impaired in the *gga1Δ gga2Δ* mutant (*Lewis et al., 2000*). The double-color 4D SCLIM observation demonstrated that Sec7-mCherry signals appeared in the vicinity of the pre-existing Tlg2p sub-compartment and then Tlg2p disappeared gradually consistent with the previous report in wild-type cells (*Figure 5C and D*, *Video 8*; *Tojima et al., 2019*). In contrast, in the *gga1Δ gga2Δ* mutant, the Sec7p sub-compartment turned over, albeit more slowly than the wild-type cells (*Figure 5—figure supplement 1E*), whereas the turnover of the Tlg2p sub-compartment was significantly delayed, resulting in an increase of the sub-compartments displaying only GFP-Tlg2p (*Figure 5E–G*, *Video 9*). No vacuolar membrane localization of GFP-Tlg2p was detected by 3D SCLIM imaging, presumably because the fluorescent signal was too weak. Line scan analyses revealed that Tlg2p and Sec7p sub-compartments were clearly segregated in the *gga1Δ gga2Δ* mutant (*Figure 5H and I*, *Video 10*). We also examined the localization of Tlg1p and found that it shows similar dynamics to Tlg2p in both wild-type and the *gga1Δ gga2Δ* cells (*Figure 5—figure supplement 2A–F*). These results suggest that the Tlg2p and Sec7p sub-compartments are discrete entities in the mutant.

Since GGA adaptors are required for the turnover of the Tlg2p sub-compartment, we speculate that the adaptors play a role there. To examine whether Gga2p is localized at the Tlg2p sub-compartment,

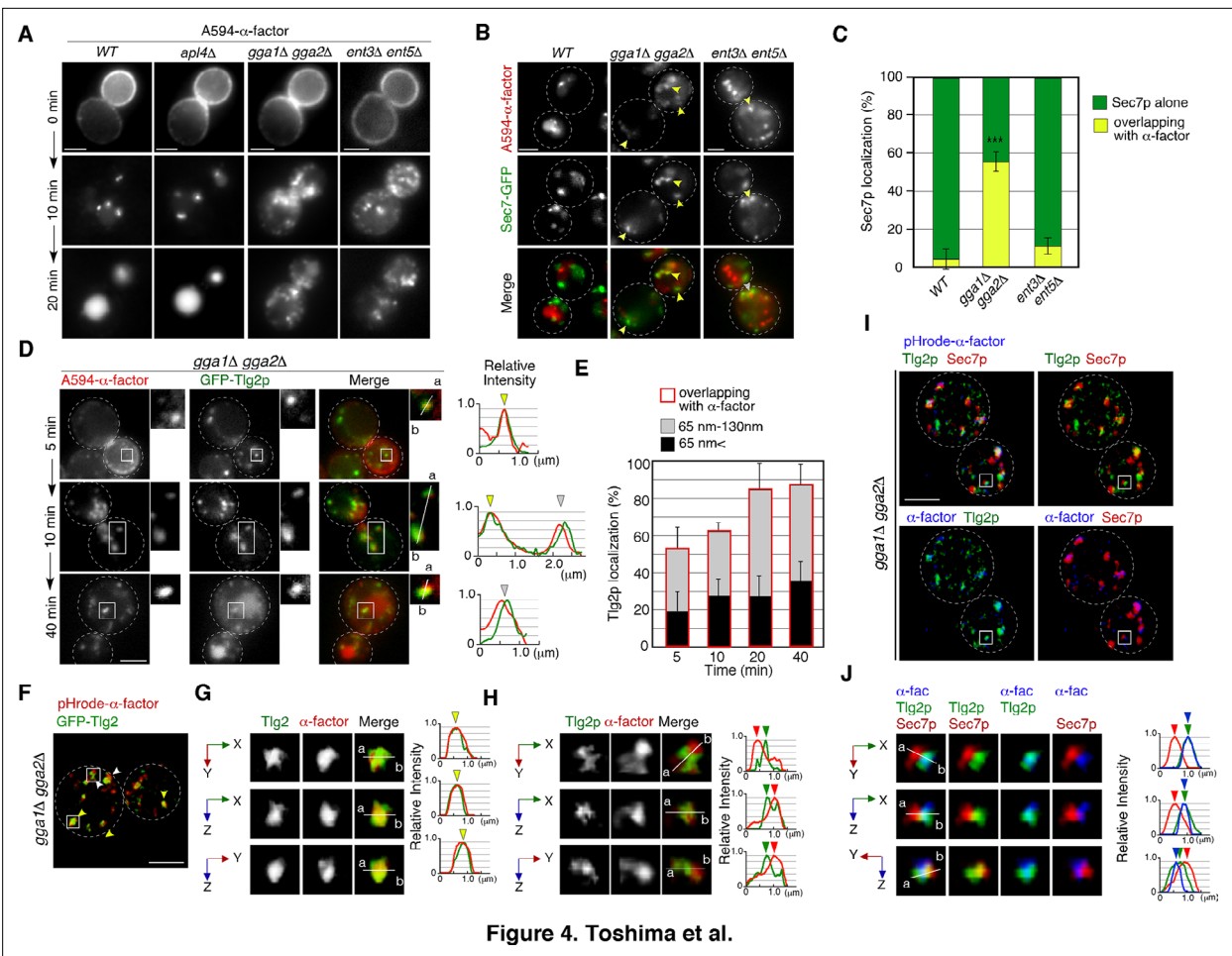

**Figure 4.** GGA adaptors are required for export of A594-α-factor out of the Tlg2p-residing compartment. (**A**) 2D imaging of A594-α-factor in cells lacking clathrin adaptor proteins. (**B**) 2D imaging of A594-α-factor and Sec7-GFP in cells lacking clathrin adaptor proteins. The images were acquired simultaneously at 20min after A594-α-factor internalization. Arrowheads indicate examples of overlapping localization. (**C**) Quantification of Sec7-GFP overlapping with A594-α-factor. Error bars indicate the SD from n ≥ 3 experiments (n > 30 puncta for each experiment). ***p<0.001, unpaired *t*-test with Welch's correction. (**D**) 2D imaging of A594-α-factor and GFP-Tlg2p in *gga1Δ gga2Δ* cells. Higher-magnification views of the boxed area are shown in the right panels. Representative fluorescence intensity profiles along lines in the merged images are indicated in the right panels. (**E**) Quantification of GFP-Tlg2p overlapping with A594-α-factor in *gga1Δ gga2Δ* cells. The bars surrounded by red lines indicate the total ratio of the Tlg2p sub-compartment overlapping with α-factor. Error bars indicate the SD from n ≥ 3 experiments (n > 30 puncta for each experiment). (**F**) 3D SCLIM imaging of GFP-Tlg2p and pHrode-α-factor in *gga1Δ gga2Δ* cells. The images were acquired simultaneously at 20min after pHrode-α-factor internalization. (**G, H**) Multi-angle magnified 3D views of the boxed areas in (**F**), representing co-localization (**G**) or adjacent localization (**H**) of GFP-Tlg2p and pHrode-α-factor. (**I**) 3D SCLIM imaging of GFP-Tlg2p, Sec7-iRFP, and pHrode-α-factor in *gga1Δ gga2Δ* cells. The images were acquired simultaneously at 10min after pHrode-α-factor internalization. (**J**) Multi-angle magnified 3D views of the boxed area in (**I**). Scale bars, 2.5μm.

The online version of this article includes the following source data for figure 4:

**Source data 1.** Data for graphs presented in *Figure 4C*.

**Source data 2.** Data for graphs presented in *Figure 4E*.

we tagged Gga2p with mCherry and observed the localization with GFP-Tlg2p. The functionality of mCherry-tagged Gga2p was confirmed by testing its ability to complement the growth phenotype of *gga1Δ gga2Δ* cells on YPD at 37°C (*Figure 5—figure supplement 2G*). Our simultaneous dual-color 2D imaging showed that~64.6% of Gga2p signals overlapped with the Tlg2p sub-compartment (*Figure 5—figure supplement 2H*). To examine the spatiotemporal localization of Gga2p precisely, we performed triple-color imaging of GFP-Tlg2p, Sec7-iRFP, and Gga2-mCherry by SCLIM. As reported previously, we observed that Gga2p appears around the Sec7p sub-compartment and disappears at a similar time as Sec7p (*Figure 5J–K*, *Video 11*; *Daboussi et al., 2012*; *Tojima et al., 2019*). As expected, a comparison of the spatiotemporal localization of Gga2p, Tlg2p, and Sec7p

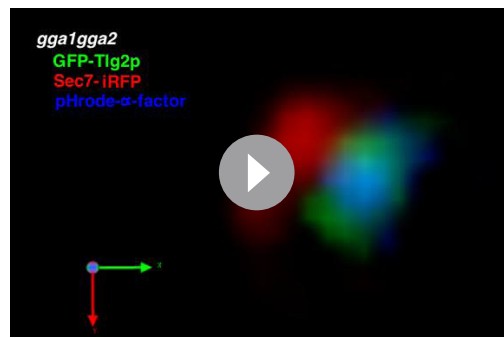

**Video 7.** Multi-angle 3D reconstructed movie of GFP-Tlg2p (green), Sec7-iRFP (red), and pHrodo-α-factor (blue) in the *gga1Δ gga2Δ* mutant.
https://elifesciences.org/articles/84850/figures#video7

in the Golgi/TGN revealed that Gga2-mCherry signals co-localize with GFP-Tlg2p-residing sub-compartments (*Figure 5K*, *Video 11*). We also found that the Gga2-mCherry signal remained after the GFP-Tlg2p signal disappeared, but the Gga2-mCherry-labeled region seemed to be distinct from Sec7-iRFP-labeled sub-compartment (*Figure 5L*, *Video 11*). Thus, Gga2p likely appears at the Tlg2p sub-compartment during the decay phase of Tlg2p and mediates the transport of α-factor and presumably Tlg2p itself from the Tlg2p sub-compartment to the Vps21p-residing endosomal compartment.

## Endocytosed Snc1p is recycled back to the PM via the Tlg2p sub-compartment

Previous studies have shown that Snc1p, a putative endocytic R-SNARE, transiently localizes to the Sec7p sub-compartment after being endocytosed and returns to the cell surface through the secretory pathway (*Best et al., 2020*; *Robinson et al., 2006*). We have demonstrated in the present study that the compartment where Snc1p's cognate Q-SNARE, Tlg2p, resides, is the first destination for endocytic traffic, and that endocytic cargo is transported to the Vps21p-residing compartment without passing through the Sec7p sub-compartment. Thus, we next wished to determine whether the pathway of Snc1p back to the PM is also mediated through the Tlg2p and Sec7p sub-compartments. To this end, we utilized fluorescent protein-tagged Snc1p, whose functionalities have been previously confirmed (*Lewis et al., 2000*). We first confirmed that GFP-Snc1p and A594-α-factor are loaded into the same endocytic vesicle. Using total internal reflection fluorescence microscopy (TIRFM), we observed that some A594-α-factor and GFP-Snc1p were localized in the same vesicles and moved together in the vicinity of the PM surface in wild-type cells (*Figure 6A*). We also observed vesicles that contained only A594-α-factor or GFP-Snc1p, perhaps because some vesicles do not contain enough A594-α-factor and some contain endogenous untagged Snc1p instead of GFP-Snc1p. As reported previously (*Black and Pelham, 2000*), GFP-Snc1p accumulated at intracellular structures in the *gga1Δ gga2Δ* mutant while it was localized at the PM and some intracellular puncta in wild-type cell (*Figure 6—figure supplement 1A*), suggesting that the transport of GFP-Snc1p to the PM, as well as that of endocytic cargo to the vacuole, is impaired. We found that the intracellular localization of GFP-Snc1p overlapped well with the Tlg2p sub-compartment in both wild-type (73.2 ± 12.4%) and *gga1Δ gga2Δ* mutant cells (78.0 ± 11.4%) (*Figure 6—figure supplement 1B*), suggesting that GFP-Snc1p is sorted to the PM through the Tlg2p sub-compartment.

We then wished to determine which sub-compartment, the Tlg2p-residing or the Sec7p-residing, mediates Snc1p transport form the TGN to the PM. Simultaneous triple-color observation by SCLIM showed that mCherry-Snc1p was localized at the PM and some intracellular puncta, similar to images obtained by 2D epi-fluorescence microscopy (*Figure 6—figure supplement 1C*). To detect the localization of Snc1p at these intracellular puncta more clearly, we reduced the fluorescent signals at the PM by contrast modulation, and found that mCherry-Snc1p was localized mostly to the sub-compartment in which GFP-Tlg2p was present (*Figure 6B*). Line scan analyses showed that peaks of mCherry-Snc1p and GFP-Tlg2p signal are almost coincident, but that the peak of Sec7-iRFP signal is slightly apart from them (*Figure 6C*). Interestingly, 4D SCLIM imaging demonstrated that the mCherry-Snc1p signal remained in the compartment from which GFP-Tlg2p departed and then disappeared at the similar timing to Sec7-iRFP (*Figure 6D*). This observation suggests that Tlg2p and Snc1p are both sorted to this compartment, but then transported by distinct trafficking pathways with different timings. The compartment in which Snc1p remains after Tlg2p leaves appeared to be distinct from the Sec7p sub-compartment (*Figure 6E*). We next confirmed the localization of Snc1p using the *gga1Δ gga2Δ* mutant because in the *gga1Δ gga2Δ* mutant the Tlg2p and Sec7p sub-compartments were more clearly segregated (*Figure 5E–I*). As expected, mCherry-Snc1p signals did not co-localize with Sec7-mCherry but clearly co-localized with GFP-Tlg2p signals (*Figure 6F and G*, *Video 12*). Time-lapse

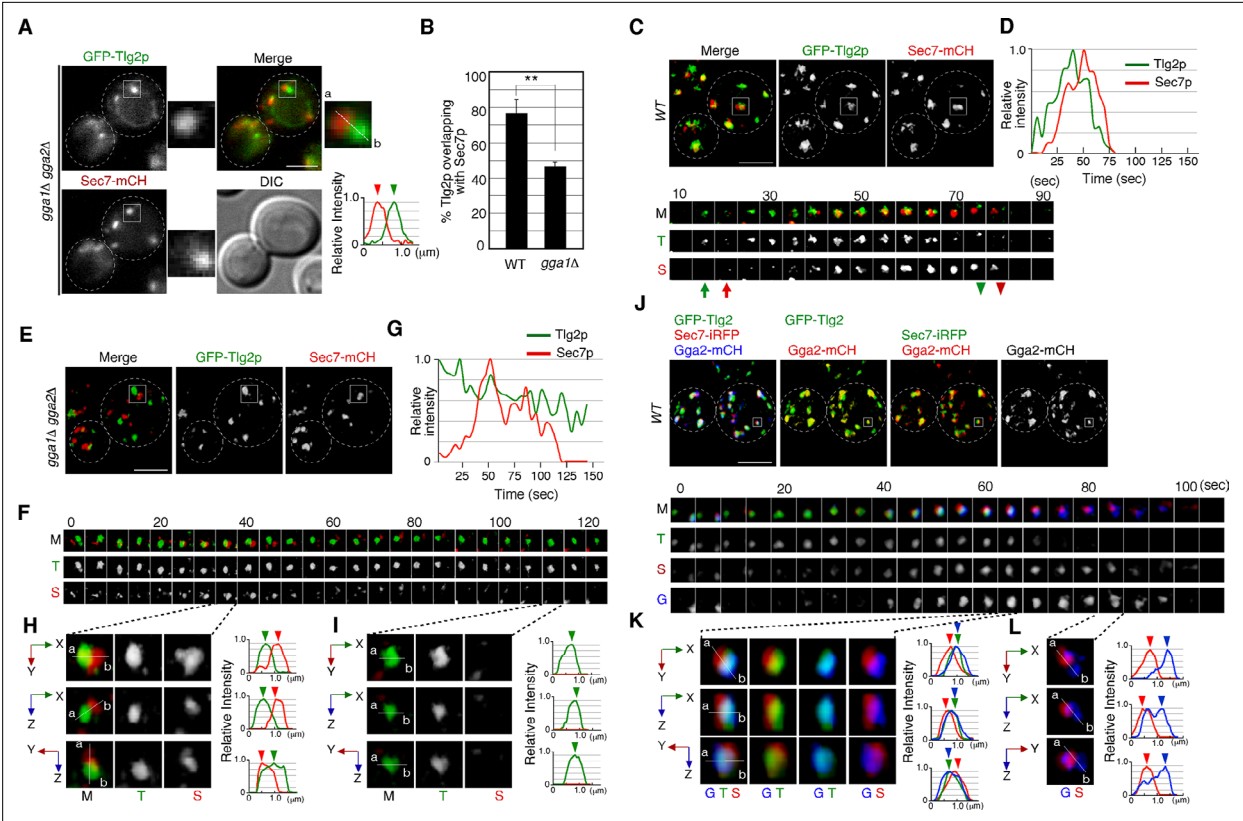

**Figure 5.** The transition from the Tlg2p- to the Sec7p-residing compartment requires GGA adaptors. (**A**) 2D imaging of GFP-Tlg2p and Sec7-mCherry in *gga1Δ gga2Δ* cells. Representative intensity profiles of GFP-Tlg2p or Sec7-mCherry along a line in the merged images are indicated in the right lower panel. (**B**) Quantification of GFP-Tlg2p overlapping with Sec7-mCherry in wild-type and *gga1Δ gga2Δ* cells. Error bars indicate the SD from n ≥ 3 experiments (n > 30 puncta for each experiment). **p<0.01, unpaired *t*-test with Welch's correction. (**C, E**) 4D super-resolution confocal live imaging microscopy (SCLIM) imaging of GFP-Tlg2p and Sec7-mCherry in wild-type cells (**C**) and *gga1Δ gga2Δ* cells (**E**).The time series of regions in the boxed areas in (**C**) are shown in the lower panels. Arrows and arrowheads denote the appearance and disappearance of each marker. (**D, G**) Time-course changes in relative fluorescence intensity of GFP-Tlg2p and Sec7-mCherry in the boxed areas in (**C**) or (**E**). (**F**) Time series of the region in the boxed area in (**E**). (**H, I**) Multi-angle magnified 3D views from (**F**). (**J**) 4D SCLIM imaging of GFP-Tlg2p, Gga2-mCherry, and Sec7-iRFP. The time series of regions in the boxed areas in (**J**) are shown in the lower panels. (**K, L**) Multi-angle magnified 3D views of time points from (**J**).Scale bars, 2.5μm.

The online version of this article includes the following source data and figure supplement(s) for figure 5:

**Source data 1.** Data for graphs presented in *Figure 5B*.

**Figure supplement 1.** Localization of Tlg2p and Sec7p at different cell cycle stages.

**Figure supplement 1—source data 1.** Data for graphs presented in *Figure 5—figure supplement 1B*.

**Figure supplement 1—source data 2.** Data for graphs presented in *Figure 5—figure supplement 1D*.

**Figure supplement 1—source data 3.** Data for graphs presented in *Figure 5—figure supplement 1E*.

**Figure supplement 2.** Dynamics of Tlg1p and Tlg2p in wild-type and *gga1Δ gga2Δ* cells.

**Figure supplement 2—source data 1.** Data for graphs presented in *Figure 5—figure supplement 2H*.

imaging showed that mCherry-Snc1p signals were also persistent with GFP-Tlg2 signals after the disappearance of Sec7-iRFP (*Figure 6H*). To further confirm that the Tlg2p sub-compartment is an early/sorting compartment, we utilized mutant lacking the *RCY1* gene because Rcy1p, a F-box protein, has been shown to be required for the transport of Snc1p to the PM (*Galan et al., 2001*; *Ma and Burd, 2020*; *Wiederkehr et al., 2000*). Previous studies have demonstrated that in the *rcy1Δ* mutant Snc1p partially co-localizes with Tlg1p but does not co-localize with Sec7p (*Best et al., 2020*; *Ma and Burd, 2019*). Similar to the case of Tlg1p, 3D SCLIM imaging demonstrated that mCherry-Snc1p signals overlap mainly with the Tlg2p sub-compartment adjacent to the Sec7p sub-compartment in the mutant (*Figure 6I*, *Video 13*). These results clearly indicate that endocytosed Snc1p is sorted to the PM via the Tlg2p sub-compartment, and taken together with the results obtained from assays

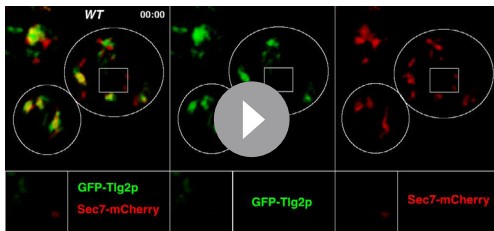

**Video 8.** Dual-color 4D movie of GFP-Tlg2p (green) and Sec7-mCherry (red) in wild-type cell. Arrows indicate examples of sequential appearance and disappearance of each protein.
https://elifesciences.org/articles/84850/figures#video8

**Video 10.** Multi-angle 3D reconstructed movie of GFP-Tlg2p (green) and Sec7-mCherry (red) in the *gga1Δ gga2Δ* mutant.
https://elifesciences.org/articles/84850/figures#video10

using fluorescently labeled α-factor, suggest that the Tlg2p-residing region serves as the endocytic early/sorting compartment.

## Yeast Rab11, Ypt31p, is localized at the Tlg2p sub-compartment as well as at the Sec7p sub-compartment

We next examined the localization of Ypt31p/32p, which are known to function in the endocytic recycling pathway (*Chen et al., 2005*). Ypt31p/32p are yeast homologues of Rab11, which localizes to recycling endosomes in mammalian cells (*Benli et al., 1996*). As it has also been reported that Ypt31p/32p are essential for export of secretory cargo from the TGN to the PM (*Jedd et al., 1997*), we investigated the sub-compartment at which Ypt31p/32p are localized. The functionality of GFP-tagged Ypt31p was confirmed by testing its ability to grow in a *ypt32Δ* background (*Figure 7—figure supplement 1A*) because deletion of both the *YPT31* and *YPT32* genes is lethal (*Jedd et al., 1997*). A previous study had demonstrated that the compartments in which Ypt31p and Ypt32p reside, and the timing of their recruitment to the TGN, are similar although localization of Ypt32p at the TGN within the bud is more prominent (*Gingras et al., 2022*). 3D SCLIM imaging also revealed that Ypt31p and Ypt32p exhibit similar localization and dynamics (*Figure 7—figure supplement 1B–D*). We next compared the localization of A594-α-factor with that of GFP-Ypt31p, and found that Ypt31p showed considerably low overlap with A594-α-factor signals at 5–20 min after A594-α-factor internalization (*Figure 7—figure supplement 1E and F*). Similar to the previous observation (*Highland and Fromme, 2021*; *Suda et al., 2013*), 4D SCLIM imaging demonstrated that Ypt31p appears at the TGN slightly later than Sec7p and disappears from the TGN after Sec7p (*Figure 7A and B*). Line scan analysis of the TGN, which includes Tlg2p, Sec7p, and Ypt31p, revealed that the mCherry-Ypt31p signal partially overlaps with both the Sec7-iRFP and GFP-Tlg2p signals (*Figure 7C*). The mCherry-Ypt31p signal remained after the Sec7-iRFP signal had disappeared, and then gradually disappeared itself (*Figure 7D*). To further examine the localization of Ypt31p/32p in the recycling pathway, we performed 3D SCLIM

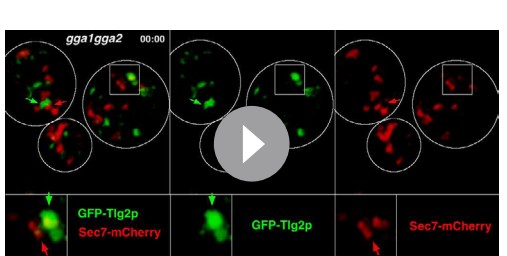

**Video 9.** Dual-color 4D movie of GFP-Tlg2p (green) and Sec7-mCherry (red) in the *gga1Δ gga2Δ* mutant.
https://elifesciences.org/articles/84850/figures#video9

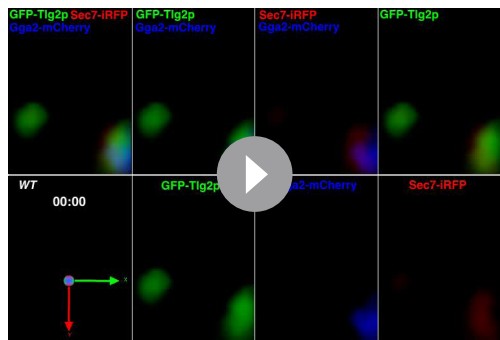

**Video 11.** Triple-color 4D movie of GFP-Tlg2p (green), Sec7-iRFP (red), and Gga2-mCherry (blue) in a wild-type cell. Arrows indicate examples of the sequential appearance and disappearance of each protein.
https://elifesciences.org/articles/84850/figures#video11

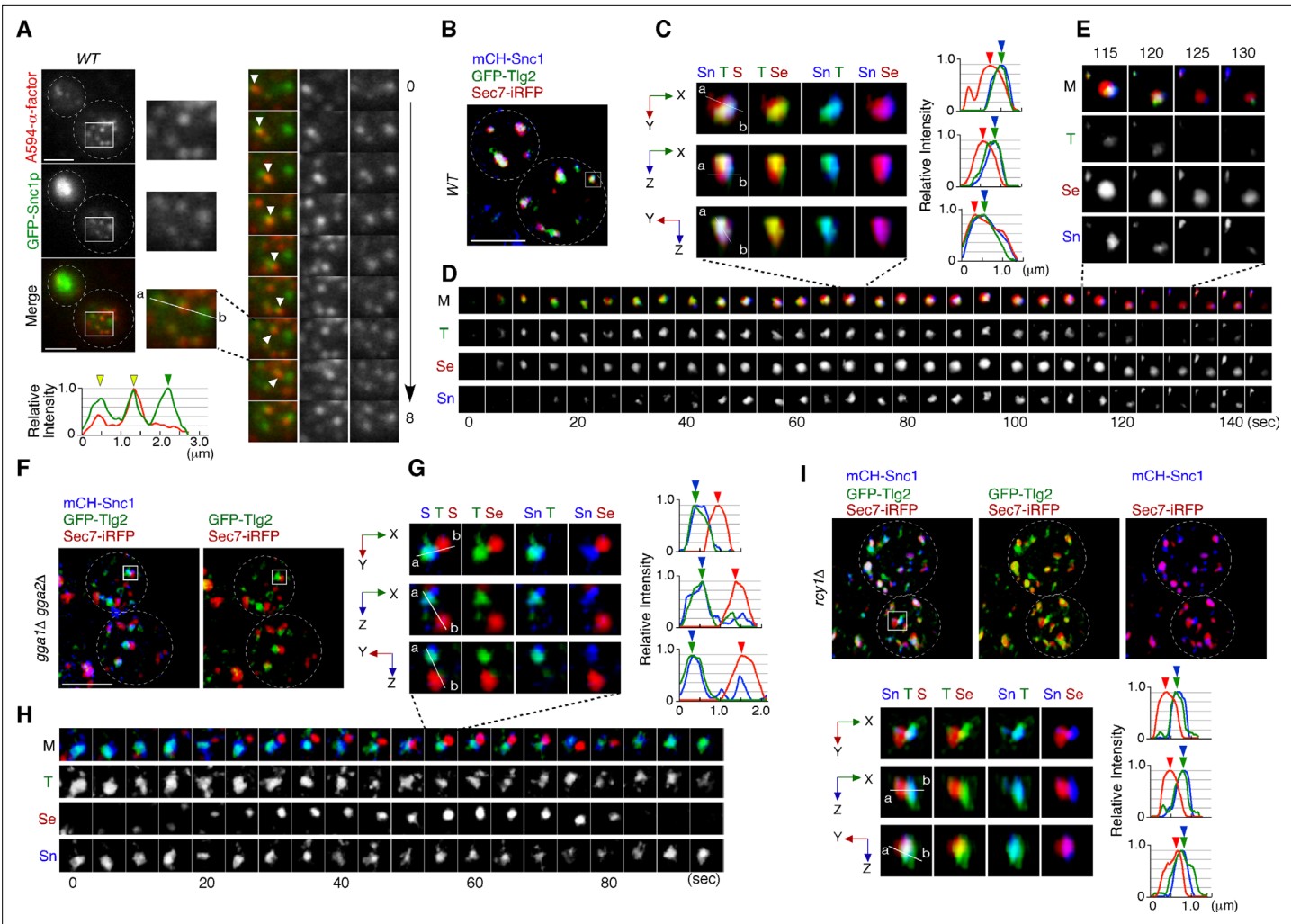

**Figure 6.** Snc1p is sorted to the plasma membrane (PM) via the Tlg2p-residing compartment. (**A**) Total internal reflection fluorescence (TIRF) imaging of GFP-Snc1p and A594-α-factor in wild-type cells. The images were acquired simultaneously at 5 min after A594-α-factor internalization. Arrowheads indicate A594-α-factor puncta including GFP-Snc1p. (**B–E**) 4D super-resolution confocal live imaging microscopy (SCLIM) imaging of GFP-Tlg2p, mCherry-Snc1p, and Sec7-iRFP in wild-type cells. (**C**) Multi-angle magnified 3D views from the 70 s image in (**D**). Representative fluorescence intensity profiles along lines (direction from 'a' to 'b') in the merged images are indicated in the right panels. (**D**) Time series of regions in the boxed area in (**B**). (**E**) Higher-magnification views of the indicated time points. (**F–H**) 4D SCLIM imaging of GFP-Tlg2p, mCherry-Snc1p and Sec7-iRFP in *gga1Δ gga2Δ* cells. (**G**) Multi-angle magnified 3D views and representative fluorescence intensity profiles at 55 s in (**H**). (**H**) The time series of region in the boxed area in (**F**). (**I**) 3D SCLIM imaging of GFP-Tlg2p, mCherry-Snc1p, and Sec7-iRFP in *rcy1Δ* cells. Multi-angle magnified 3D views of the boxed area and representative fluorescence intensity profiles shown in the lower panels. Scale bars, 2.5 μm.

The online version of this article includes the following source data and figure supplement(s) for figure 6:

**Figure supplement 1.** Localization of Snc1p at the Tlg2p-residing compartment.

**Figure supplement 1—source data 1.** Data for graphs presented in *Figure 6—figure supplement 1B*.

imaging and found that the peaks of mCherry-Snc1p and Sec7-iRFP were slightly apart, but that the GFP-Ypt31p signal entirely overlapped them (*Figure 7E–G*). Additionally, 4D SCLIM imaging demonstrated that the GFP-Ypt31p signal remained in the compartment where mCherry-Snc1p was present after Sec7p had disappeared (*Figure 7F and H*). These observations suggest that Ypt31p resides in both the Tlg2p and Sec7p compartments, and presumably functions in the secretory and recycling pathways. We also examined Ypt31p localization in the *gga1Δ gga2Δ* mutant. As shown in *Figure 6G and H*, the mCherry-Snc1p and Sec7-iRFP signals were clearly segregated in the *gga1Δ gga2Δ* mutant, and mCherry-Snc1p signals persisted after the disappearance of Sec7-iRFP (*Figure 7J and L*). Interestingly, the timing of the appearance and disappearance of GFP-Ypt31p at the TGN

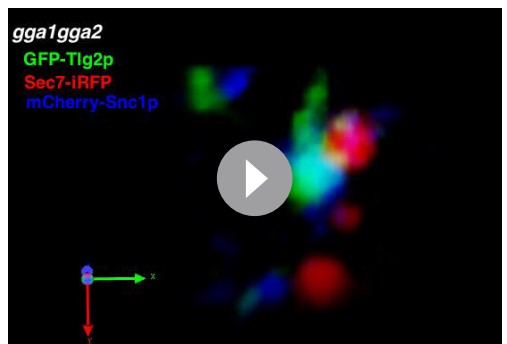

**Video 12.** Multi-angle 3D reconstructed movie of GFP-Tlg2p (green), Sec7-iRFP (red), and mCherry-Snc1p (blue) in the *gga1Δ gga2Δ* mutant.
https://elifesciences.org/articles/84850/figures#video12

in the mutant was similar to that in the wild-type cell, but the GFP-Ypt31p signal was observed only in the compartment where Sec7p resides (*Figure 7J–L*). Thus, recruitment of Ypt31p to the Tlg2p sub-compartment appeared to be impaired in the *gga1Δ gga2Δ* mutant.

## Discussion

### The Tlg2p sub-compartment functions as an endocytic early/sorting compartment

On the basis of the data presented here and in previous studies, we propose a role for the Tlg2p sub-compartment as an early/sorting compartment in the endocytic pathway (*Figure 8*). A previous observation showed that the TGN where Sec7p resides is the first destination for endocytic traffic (*Day et al., 2018*). Spatially, the Sec7p-residing region is located so closely to the Tlg2p-residing region that previously they were not considered to be distinct compartments. In the present study, we have shown that the Tlg2p-residing structure exists as a sub-compartment distinct from the Sec7p-residing structure within the TGN, thereby revealing that endocytic cargo is incorporated into the Tlg2p sub-compartment within the TGN. Consistent with this result, previous studies using electron microscopy have shown that endocytic cargo is first transported to the tubular/vesicular structure that contains Tlg1p after internalization (*Prescianotto-Baschong and Riezman, 1998*). We have also shown that Snc1p, the endocytic R-SNARE that targets Tlg2p, is transiently localized to this sub-compartment and exits from there to the PM, whereas Tlg2p is transported to the PVC via the Vps21p-residing compartment. As shown in *Figure 6*, Tlg2p appears at the TGN earlier than Snc1p and disappears from the TGN before Snc1p. This suggests that Snc1p binds to Tlg2p that has been recycled back from the PVC and dissociates from Tlg2p at the sub-compartment. Recent studies have revealed that yeast has a direct pathway that recycles cell surface protein back from the endosome to the PM (*Laidlaw et al., 2022*). In this pathway, unlike Snc1p, cargos are not transported to the TGN, but are incorporated directly into the Vps4p-residing endosome and returned back to the PM (*Adell et al., 2017*; *Laidlaw et al., 2022*). Since GFP-Snc1p is known to little overlap with Vps4p-residing endosomes (*Laidlaw et al., 2022*), the Tlg2p sub-compartment revealed in this study is likely to be distinct from the Vps4p-residing endosome.

The recycling of Snc1p is mediated by several factors, including Rcy1p and Ypt31/32p (*Chen et al., 2005*; *Furuta et al., 2007*). Rcy1 forms a complex with the GFP-bound form of Ypt31/32 GTPases (*Chen et al., 2005*) and associates with the phosphatidylserine flippase Drs2-Cdc50, interaction with which is required for PM recycling of Snc1p (*Furuta et al., 2007*; *Hanamatsu et al., 2014*). Previous studies have demonstrated that Snc1p accumulates in Tlg1p-residing puncta distinct from Sec7p-residing TGN in the *rcy1Δ* or *cdc50*-ts mutant (*Best et al., 2020*; *Furuta et al., 2007*; *Ma and Burd, 2019*). In the present study, we have shown that Ypt31p resides in both the Tlg2p and Sec7p compartments, and presumably functions in the secretory and recycling pathways (*Figure 8*). In the *gga1Δ gga2Δ* mutant, localization of Ypt31p was only observed in the Sec7p sub-compartment that turned over relatively normally, and recruitment of Ypt31p to the Tlg2p

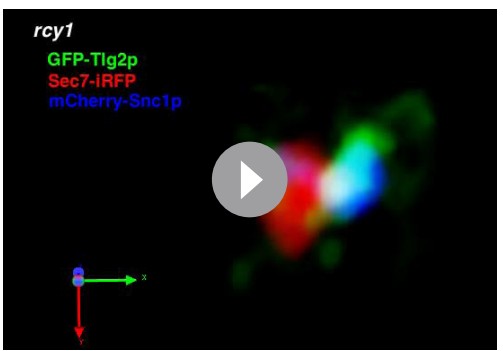

**Video 13.** Multi-angle 3D reconstructed movie of GFP-Tlg2p (green), Sec7-iRFP (red), and mCherry-Snc1p (blue) in the *rcy1Δ* mutant.
https://elifesciences.org/articles/84850/figures#video13

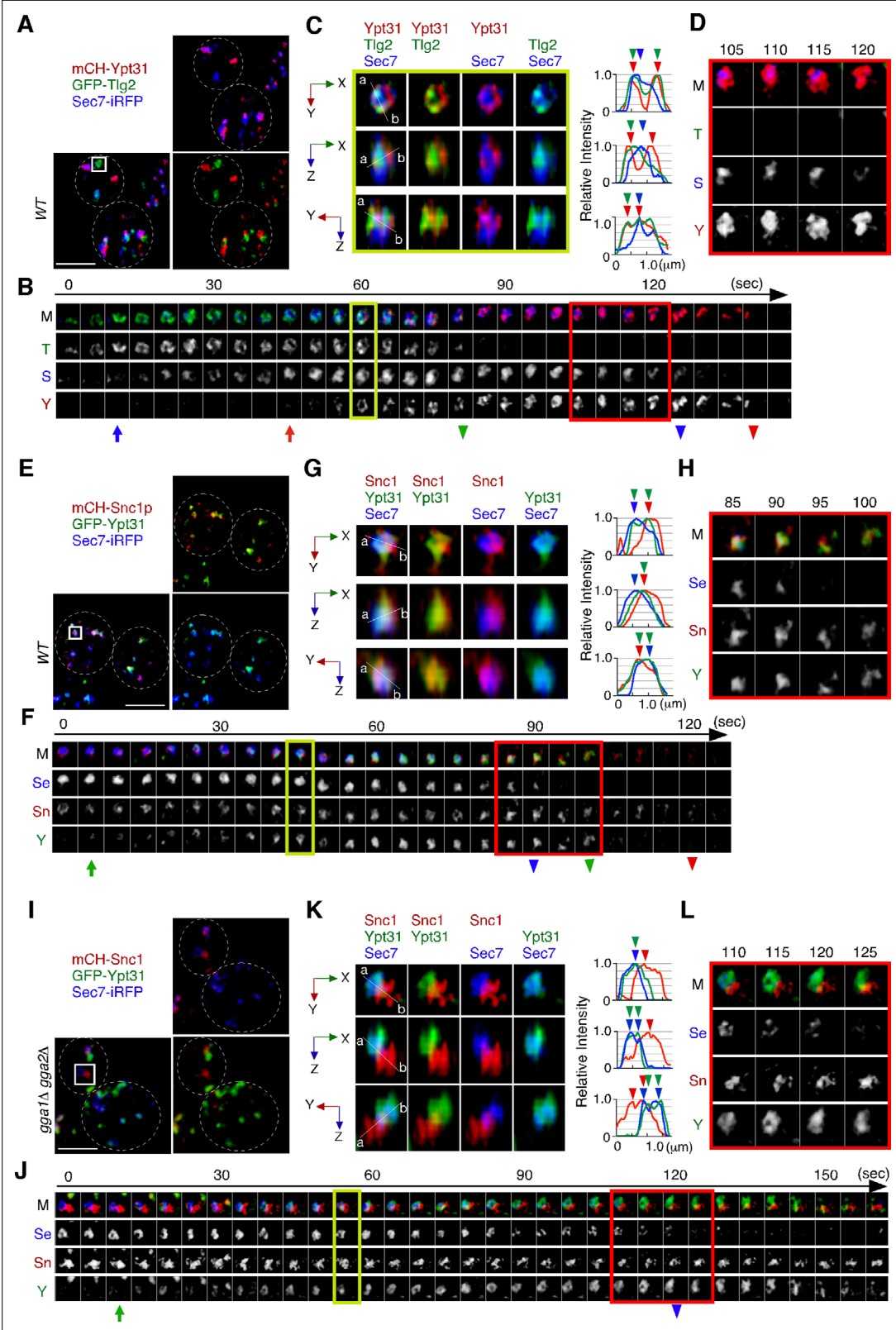

**Figure 7.** Ypt31p is localized at both the Tlg2p-residing compartment and the Sec7p-residing compartment. (**A–D**) 4D super-resolution confocal live imaging microscopy (SCLIM) imaging of GFP-Tlg2p, mCherry-Ypt31p, and Sec7-iRFP in wild-type cells. (**B**) Time series of regions in the boxed area in (**A**). Arrows and arrowheads denote the appearance and disappearance of each marker. (**C**) Multi-angle magnified 3D views from the 60 s image in (**B**). Representative fluorescence intensity profiles along lines (direction from 'a' to 'b') in the merged images are indicated in the right panels. (**D**) Higher-

*Figure 7 continued on next page*

*Figure 7 continued*

magnification views of the red-boxed area in (**B**). (**E–H**) 4D SCLIM imaging of GFP-Ypt31p, mCherry-Snc1p and Sec7-iRFP in wild-type cells. (**F**) Time series of regions in the boxed area in (**E**). Arrows and arrowheads denote the appearance and disappearance of each marker. (**G**) Multi-angle magnified 3D views from the 45 s image in (**F**). Representative fluorescence intensity profiles along lines (direction from 'a' to 'b') in the merged images are indicated in the right panels. (**H**) Higher-magnification views of the red-boxed area in (**F**). (**I–L**) 4D SCLIM imaging of GFP-Ypt31p, mCherry-Snc1p, and Sec7-iRFP in *gga1Δ gga2Δ* cells. (**J**) Time series of regions in the boxed area in (**I**). Arrows and arrowheads denote the appearance and disappearance of each marker. (**K**) Multi-angle magnified 3D views from the 55 s image in (**J**). Representative fluorescence intensity profiles along lines (direction from 'a' to 'b') in the merged images are indicated in the right panels. (**L**) Higher-magnification views of the red-boxed area in (**J**). Scale bars, 2.5 μm.

The online version of this article includes the following source data and figure supplement(s) for figure 7:

**Figure supplement 1.** Localization of Ypt31p, Ypt32p, and α-factor in wild-type cell.

**Figure supplement 1—source data 1.** Data for graphs presented in *Figure 7—figure supplement 1F*.

sub-compartment appeared to be impaired, suggesting that the functions of Ypt31p in the secretory and recycling pathways are separable.

We recently reported that Vps9p, a GEF for Vps21p, is recruited to the TGN, and then transported to the endosome to activate Vps21p through Ent3p/5p-mediated vesicle transport (*Nagano et al., 2019*). Previous studies reported that GGA adaptors-enriched vesicles include Ent3p and Ent5p (*Daboussi et al., 2012*), and thus Vps9p might be recruited to and transported from the Tlg2p subcompartment. In *ent3Δ ent5Δ* cells as well as *vps21Δ ypt52Δ* cells, Vps21p-mediated endosomal transport is impaired (*Nagano et al., 2019*), and thus α-factor and Tlg2p presumably accumulate at the Vps21p-residing compartment. In contrast, deletion of Gga1p/2p has a negligible effect on Vps21p-mediated endosome formation (*Nagano et al., 2019*), but affects turnover of the Tlg2p subcompartment (*Figure 8*).

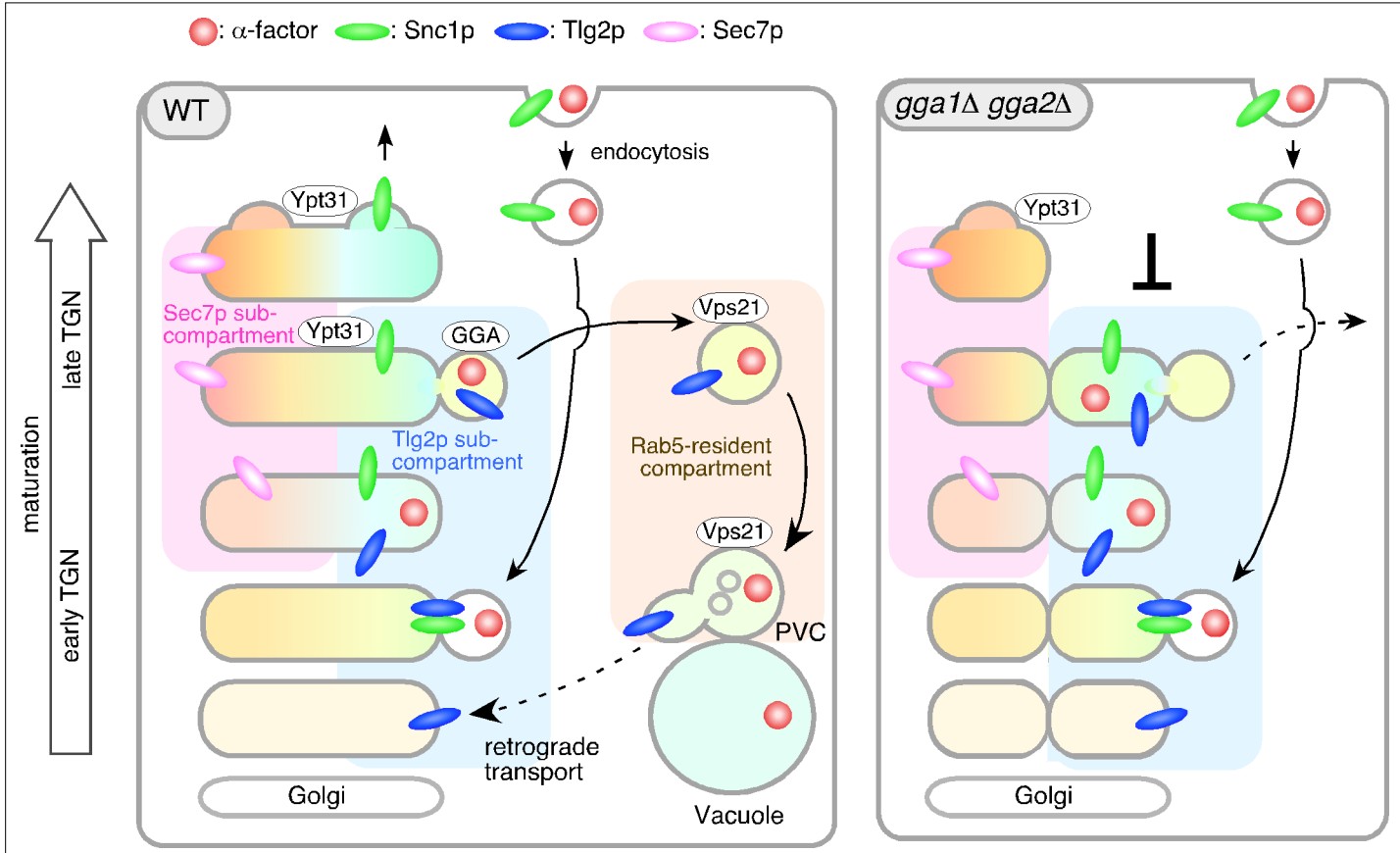

**Figure 8.** Model showing the role of the Tlg2p-residing compartment as an early/sorting compartment in the endocytic pathway. Schematic showing the Tlg2p-residing area as a discrete early/recycling sub-compartment that sorts endocytic cargo to the endocytic or recycling pathway. The effects of *gga1Δ gga2Δ* cells on the post-TGN trafficking pathway are shown on the right. See details in the text.

## Integration of the present model with earlier observations

The yeast Q-SNAREs Tlg1p and Tlg2p were first identified a few decades ago, and their roles have been debated because of their localization at the TGN and endosomal compartment (*Abeliovich et al., 1998*; *Holthuis et al., 1998a*; *Séron et al., 1998*). A previous study by Day et al. reported that both Tlg1p and Tlg2p shows substantial co-localization with Sec7p (~50% GFP-Tlg1p-labeled puncta overlap with Sec7p-residing TGN) (*Day et al., 2018*). Holthuis et al. reported that upon subcellular fractionation the two proteins peaked at different densities, leading the authors to propose that the two proteins are found in a putative early endosome, and the Golgi/PVC, respectively (*Holthuis et al., 1998a*). However, their distribution within the fractions entirely overlapped and Tlg2p is also found in the peak fraction of Tlg1p (*Holthuis et al., 1998a*). Matching these observations, we observed that both Tlg1p and Tlg2p substantially localize at the TGN while Tlg1p localizes at the PVC less than Tlg2p. Thus, Tlg1p seems to localize at the Tlg2p sub-compartment and function together with it.

By examining the localization, we demonstrated that Alexa-α-factor, transported to the Tlg2p sub-compartment adjacent to the Sec7p sub-compartment, moves to the Vps21p-residing compartment presumably together with Tlg2p. As Tlg2p interacts with Snc2p, a potential endocytic R-SNARE, and its deletion caused a defect in the endocytic pathway (*Abeliovich et al., 1998*; *Paumet et al., 2001*; *Séron et al., 1998*), the primary role of Tlg2p in the endocytic pathway seems to be in mediating fusion of the endocytic vesicle with the early/sorting compartment. Additionally, Tlg2p present in the Vps21p-residing compartment might have a role in endosomal fusion. After being transported to the Vps21p-residing compartment, Tlg2p may be returned to the early/sorting compartment by interacting with Ypt6p (yeast Rab6) and the Golgi-associated retrograde protein (GARP) complex (*Siniossoglou and Pelham, 2001*; *Suda et al., 2013*).

Tojima et al. have previously demonstrated that Tlg2p and Sec7p overlap during Golgi/TGN maturation (*Tojima et al., 2019*), and Day et al. have reported that the Sec7p-residing TGN serves as an early/sorting endosome, like plant cells (*Day et al., 2018*). Our present analysis indicates that Tlg2p partially overlaps with Sec7p, although the major part of the Tlg2p sub-compartment exists separate from the Sec7p-residing one. The clathrin-residing region has also been shown to segregate from the Tlg2p sub-compartment, although it largely overlaps with the Sec7p sub-compartment (*Tojima et al., 2019*), supporting the idea that the Tlg2p and Sec7p sub-compartments are segregated spatially. Our model, therefore, is not inconsistent with those built on previous observations, but adds the concept that the Tlg2p-residing region, rather than the Sec7p-residing region, functions as an endocytic early/sorting compartment. In contrast, Best et al. proposed a model in which yeast has a Tlg1p-residing endosome distinct from the TGN, and that this mediates Snc1p recycling via three different routes by Drs2/Rcy1/COPI, Snx4-Atg20, and the retromer (*Best et al., 2020*). Their observation that Snc1p is mis-localized to the Tlg1p-residing compartment in the *rcy1Δ* cell is in agreement with our result (*Best et al., 2020*). An Snx4p-dependent route facilitating retrieval of Snc1p from the PVC to the Sec7p-residing TGN is also not inconsistent with our result if the retrieval region is the Tlg2p sub-compartment rather than the Sec7p sub-compartment. Thus, we can consider their model to be compatible with ours, given that the Tlg2p- and Sec7p-residing regions are adjacent but independent sub-compartments within the TGN.

## Endocytic recycling compartments in different organisms

The identity of the sorting/recycling compartment (SE/RE) varies among different organisms (*Nakano, 2022*). In plants, the TGN serves as the SE/RE, whereas in mammalian cells the SE/RE functions as a different organelle distinct from the TGN (*Grant and Donaldson, 2009*). A recent study has reported that, in *Drosophila* and microtubule-disrupted HeLa cells, recycling endosomes (REs) present two distinct stages – Golgi-associated REs and Golgi-independent REs – that are interconvertible (*Fujii et al., 2020a*). Plants have two types of TGN – Golgi-associated TGN and Golgi-independent-TGN (*Kang et al., 2011*; *Uemura et al., 2014*; *Viotti et al., 2010*) – whose dynamics are similar to REs in *Drosophila* and mammalian cells (*Fujii et al., 2020a*; *Fujii et al., 2020b*). These observations suggest that the properties of *Drosophila* and mammalian RE are similar to those of plant cells. In this study, we have shown that yeast has a sorting compartment within the TGN and that this structure seems to be more similar to that of a plant cell. A recent study has demonstrated that in plants the TGN has at least two subregions (zones) responsible for secretory and vacuolar trafficking (*Nakano, 2022*; *Shimizu et al., 2021*), and that the yeast sorting/recycling compartment might correspond to the latter zone.

Interestingly, in *Drosophila* cells markers of the TGN and RE show close localization and even partial overlap (*Fujii et al., 2020a*), and these structures appear to have some similarity to the yeast sorting/recycling compartment.

## Possible mechanism for early/sorting compartment generation

How the endocytic early/sorting compartment is generated and turned over has been unclear both in plants and mammals. Our findings provide a new insight into the mechanism. Recent studies have demonstrated that the appearance and the dynamic behaviors of Golgi/TGN-resident proteins exhibit a unique order of events during Golgi/TGN maturation (*Kim et al., 2016*; *Kurokawa et al., 2019*; *Thomas et al., 2021*; *Tojima et al., 2019*). Tojima et al. proposed that the Golgi/TGN maturation process can be classified into three successive stages: the Golgi stage, the early TGN stage, and the late TGN stage; Tlg2p was categorized as an early TGN-resident protein (*Tojima et al., 2019*). Our present results indicate that, after capturing endocytic vesicles, the Tlg2p sub-compartment gradually disappears through export of Tlg2p to the Vps21p-residing compartment. The Tlg2p sub-compartment is presumably regenerated by Sys1p-mediated retrograde transport. Sys1p, a late Golgi-resident protein, recruits GARP complexes through the Sys1-Arl3-Arl1-Imh1 cascade (*Graham, 2004*) and interacts with vesicles containing Tlg1p derived from endosomes (*Chen et al., 2019*; *Siniossoglou and Pelham, 2001*). Tlg1p is known to bind directly to the GARP complex and mediates the retrograde transport from the late endosome (*Siniossoglou and Pelham, 2001*). Tlg2p has been also reported to be cycled between the TGN and endosome (*Lewis et al., 2000*). Interestingly, a previous study reported that the majority of the Sys1p-residing compartment is not accompanied by clathrin appearance (*Tojima et al., 2019*), suggesting that Sys1p resides in a distinct compartment from the Sec7p-residing one. The observation that the Sys1p-residing compartment appears before Tlg2p and matures into the Tlg2p-residing one (*Tojima et al., 2019*) supports the idea that Sys1p may play a role in generating the early/sorting compartment.

## Role of GGA adaptors in endocytic cargo transport

A previous study has suggested the importance of GGA adaptors in the export of α-factor from the Sec7p sub-compartment (*Day et al., 2018*). Black and Pelham reported that Pep12p, a yeast syntaxin localized primarily at the late endosome, is also mislocalized to the Tlg1p high-density membranes in *gga1Δ gga2Δ* cell and suggested that an aberrant early endosome structure may cause a defect in the TGN-to-endosome traffic (*Black and Pelham, 2000*). Here we have shown that in *gga1Δ gga2Δ* cells α-factor accumulates at the Tlg2p sub-compartment. Interestingly, in *gga1Δ gga2Δ* cells, turnover of the Tlg2p- and Tlg1p-residing sub-compartment is impaired, and these observations suggest that export of cargos by GGA adaptors is important for normal turnover of the Tlg2p sub-compartment. As GGA adaptors are reported to bind directly to ubiquitin, which function as a sorting signal for lysosomal degradation (*Scott et al., 2004*), the ubiquitination signal could mediate the export of α-factor from the Tlg2p sub-compartment. Several cell-surface proteins, including the α-factor receptor Ste3p, are known to be ubiquitinated, thereby being sorted from the TGN to the vacuole (*Buelto et al., 2020*; *Scott et al., 2004*). The α-factor receptor Ste2p is also ubiquitinated upon ligand binding, promoting incorporation of the α-factor-Ste2p complex into the clathrin-coated vesicle (*Hicke and Riezman, 1996*; *Toshima et al., 2009*). Thus, after being transported to the Tlg2p sub-compartment, the ubiquitinated α-factor-Ste2p complex could be exported to the Vps21p-residing compartment by binding to GGA adaptors.

Two major clathrin adaptors, GGA adaptors and the AP-1 complex, were implicated to act in TGN-endosome trafficking (*Traub, 2005*). Daboussi et al. showed that the major population of Gga2p and AP-1 is separated both temporally and spatially, and that Gga2p arrives earlier than AP-1 at almost the same time point as Sec7p (*Daboussi et al., 2012*). In contrast, Gga1p appears to arrive at the TGN earlier than Gga2p, presumably with similar timing to Tlg2p because it arrives 3 s earlier than Chs5p, a component of the exomer complex (*Anton-Plagaro et al., 2021*). A recent study has reported that GGA adaptors, but not the AP-1 complex, are necessary for the transport of newly synthesized vacuolar protein from the TGN to the vacuole via the VPS pathway (*Casler and Glick, 2020*). We previously demonstrated that convergence of the endocytic and VPS pathways occurs upstream of the requirement for Vps21p in the early stage of the endocytic pathway (*Toshima et al., 2014*). Taken together, these observations suggest that the endocytic cargo derived from the PM and

the biosynthetic cargo for the VPS pathway both reach the Tlg2p sub-compartment and exit there by a GGA-adaptor-dependent mechanism.

## Materials and methods

### Yeast strains, growth conditions, and plasmids

The yeast strains used in this study are listed in *Supplementary file 1*. All strains were grown at 25°C in standard rich medium (YPD) or synthetic medium (SM) supplemented with 2% glucose and appropriate amino acids. The N-terminal GFP-tagged Tlg2p was expressed as follows: the SacI and HindIII fragment (containing iGFP-TLG2) extracted from YLplac211-iGFP-TLG2 plasmid (Addgene #105262) was inserted into the SacI and EcoRV-digested pRS303 (pRS303-iGFP-Tlg2). To integrate pRS303-iGFP-TLG2 into the HIS3 locus, the plasmid was linearized by NheI and transformed into wild-type or mutant cells. The N-terminal GFP tagging of Vps21p and the C-terminal fluorescent protein tagging of proteins was performed as described previously (*Toshima et al., 2014*).

### Fluorescence labeling of α-factor and endocytosis assays

Fluorescence labeling of α-factor was performed as described previously (*Toshima et al., 2006*). For endocytosis assays, cells were grown to an OD600 of ~0.5 in 0.5 ml YPD, briefly centrifuged, and resuspended in 20 µl SM with 5 µM Alexa Fluor-labeled α-factor. After incubation on ice for 2 hr, the cells were washed with ice-cold SM. Internalization was initiated by addition of SM containing 4% glucose and amino acids at 25°C.

### Fluorescence microscopy and image analysis

2D imaging was performed using an Olympus IX83 microscope equipped with a ×100/NA 1.40 (Olympus) or a ×100/NA 1.49 (Olympus) objective and Orca-AG cooled CCD camera (Hamamatsu), using Metamorph software (Universal Imaging). For TIRF illumination, optically pumped semiconductor laser (OPSL) (Coherent) with emission of at 488 nm (OBIS 488LS-50) and at 561 nm (OBIS 561LS-50) were used to excite GFP or mCherry/Alexa594, respectively. Simultaneous imaging of red and green fluorescence was performed using an Olympus IX83 microscope, described above, and an image splitter (Dual-View; Optical Insights) that divided the red and green components of the images with a 565 nm dichroic mirror and passed the red component through a 630/50 nm filter and the green component through a 530/30 nm filter. These split signals were taken simultaneously with one CCD camera, described above. 2D triple-color imaging were performed using an Olympus IX81 microscope equipped with a high-speed filter changer (Lambda 10-3; Sutter Instruments) that can change filter sets within 40ms. All cells were imaged during the early- to mid-logarithmic phase. Images for analysis of co-localization of red and green signals were acquired using simultaneous imaging (64.5 nm pixel size), described above. Intensity profiles of GFP-fused protein and mCherry-fused protein or A594-α-factor were generated using the Plot Profile tool (ImageJ v1.53a) across the center of fluorescence signals. All the data which shows two peaks of GFP and mCherry/Alexa Fluor intensity with a distance of less than 2 pixels, are defined as 'overlapping'.

3D and 4D imaging were performed with SCLIM (*Kurokawa and Nakano, 2020*; *Tojima et al., 2023*). The system is composed of an Olympus IX73 microscope, solid-state lasers with emission at 473 nm (Blues, 50 mW; Cobolt), 561 nm (Jive, 50 mW; Cobolt), and 671 nm (CL671-100-S, 100 mW; CrystaLaser), a UPlanXApo ×100/NA 1.45 (Olympus) objective, a high-speed spinning-disk confocal scanner (Yokogawa Electric), a custom-built piezo actuator (Yokogawa Electric), a custom-built emission splitter unit, image intensifiers (Hamamatsu) with a custom-made cooling system, a magnification lens system to provide ×266.7 final magnification, and three EM-CCD cameras (Hamamatsu) for green, red, and infrared fluorescence channels. The pixel size corresponds to 0.06 µm on the sample plane. For 3D observations, we collected 21 optical sections spaced 0.2 µm apart (z-range = 4.0 µm). Z-stack images were reconstructed to 3D images and deconvoluted by using theoretical point spread functions with Volocity (Quorum Technologies).

## Acknowledgements

This work was supported by JSPS KAKENHI grant #18K062291, and the Takeda Science Foundation to JYT., as well as JSPS KAKENHI grant #19K065710, the Takeda Science Foundation, and Life Science Foundation of Japan to JT.

## Additional information

### Funding

| Funder | Grant reference number | Author |
| --- | --- | --- |
| Japan Society for the Promotion of Science | #18K062291 | Junko Y Toshima |
| Japan Society for the Promotion of Science | #19K065710 | Jiro Toshima |
| Takeda Science Foundation | | Junko Y Toshima Jiro Toshima |
| Life Science Foundation of Japan | | Jiro Toshima |

The funders had no role in study design, data collection and interpretation, or the decision to submit the work for publication.

### Author contributions

Junko Y Toshima, Conceptualization, Data curation, Formal analysis, Funding acquisition, Investigation, Visualization, Writing - original draft, Project administration; Ayana Tsukahara, Makoto Nagano, Data curation; Takuro Tojima, Akihiko Nakano, Methodology, Writing – review and editing; Daria E Siekhaus, Writing – review and editing; Jiro Toshima, Conceptualization, Supervision, Funding acquisition, Writing - original draft, Project administration, Writing – review and editing

### Author ORCIDs

Junko Y Toshima http://orcid.org/0000-0001-9325-8622
Makoto Nagano http://orcid.org/0000-0002-1948-1591
Daria E Siekhaus http://orcid.org/0000-0001-8323-8353
Akihiko Nakano http://orcid.org/0000-0003-3635-548X
Jiro Toshima http://orcid.org/0000-0003-3264-9843

### Decision letter and Author response

Decision letter https://doi.org/10.7554/eLife.84850.sa1
Author response https://doi.org/10.7554/eLife.84850.sa2

## Additional files

### Supplementary files
- Supplementary file 1. List of strains used in this study.
- MDAR checklist

### Data availability

All data generated or analysed during this study are included in the manuscript and supporting file; Source Data files have been provided for *Figures 1–5* and Figure 2-figure supplementary 1B,D,E, Figure 5-figure supplementary 1H, Figure 6-figure supplementary 1B, Figure 7-figure supplementary 1F.

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
