## [Editor Report]

In this study, the authors use high-speed and high-resolution imaging to investigate the role of the yeast syntaxin homolog Tlg2p in endocytic vesicle sorting. They obtain compelling data to show that the Tlg2p-residing compartment within the trans-Golgi network functions as an early/sorting compartment, where endocytic cargos are sorted to either the recycling pathway or the endo-lysosomal pathway. The authors also describe additional molecular details of this sorting process and overall provide important insights into the mechanism of endocytic vesicle sorting in budding yeast.

---

## [Decision Letter]

**Decision letter after peer review:**

Thank you for submitting your article "The yeast endocytic early/sorting compartment exists as an independent sub-compartment within the *trans*-Golgi network" for consideration by *eLife*. Your article has been reviewed by 3 peer reviewers, and the evaluation has been overseen by a Reviewing Editor and Vivek Malhotra as the Senior Editor. The reviewers have opted to remain anonymous.

Essential revisions:

As you will see, the reviewers appreciate the quality of your imaging data, and also the significance and importance of the question that you are addressing. However, they also make quite a lot of specific comments and suggestions. Most of these can be addressed with clarifications in the text or more quantitation. There is a specific request for examining further markers such as Ypt31 or Ypt32 (Rab11) and FM-464 so as to clarify the identity of the structures that you observe. This seems reasonable. One reviewer suggests electron microscopy to examine the compartments, but we agreed that whilst this would be ideal, it would be very technically challenging in this system and so beyond the scope of what you have done. Thus the electron microscopy is not required.

One aspect that is essential to address is to discuss in more depth how your findings relate to the previous studies on this topic. There are several studies that present evidence for endocytic compartments that lie upstream of the Sec7 compartment on the endocytic pathway, and this needs to be properly discussed. The reviewers have indicated the specific papers that need to be covered. This is not a requirement to reach a definitive conclusion about all aspects of the pathway, but rather to fairly acknowledge the views in the field, and summarise what is agreed and what remains to be resolved.

*Reviewer #1 (Recommendations for the authors):*

The article is very well written and the findings are well presented. I have the following suggestions:

1. Because the yeast TGN has been suggested to function not only as an early endosome, but also as the recycling endosome, the study feels incomplete without an analysis and clear discussion of the recycling endosome. The authors should try to place the recycling endosome as well in their model of how the TGN relates to endocytosis by conducting a similar analysis of Rab11 or other markers in relation to Sec7, Tlg2, Snc1 and endocytosed/recycled cargo.

2. I would suggest not to use the cyan/green color combination to make the point that markers overlap, as this could lead to visual overestimation of that overlap (e.g. Figure 1F, 2J, 3G, 4J, 5K, 6C).

3. Line 57: "Additionally, we have previously demonstrated that endocytosed cargos are rarely transported to the Sec7p-residing TGN compartment (Toshima et al., 2014)." This needs some better description and context, as the present study shows that endocytosed cargo travels to the Tlg2 sub-compartment, which overlaps with the Sec7 sub-compartment. Should the authors talk about "Tlg2-only" and "Sec7-only" sub-compartments? However, see next point.

4. Line 140: "In contrast, Sec7-GFP, which was also detected as several punctate structures, did not show significant overlap with A594-α-factor signals either at 5 min (26.0{plus minus}9.4%) or 10 min (22.7{plus minus}6.0%) after A594-α-factor internalization (Figure 1B; Figure S1A)." In what sense are 26.0 % and 22.7 % overlap between Sec7 and α-factor "not significant"?

5. In the Discussion section called "Integrating the present model with earlier observations", the authors should discuss the findings on Snc1/Tlg1 in Best et al., 2020, and how they can be integrated with those of Day et al., 2018 and their own.

6. There is no attempt to put the findings in an evolutionary context. With the exception of some references to their own work in plants, there is no discussion of the implications for other organisms, such as *Drosophila* or humans.

*Reviewer #2 (Recommendations for the authors):*

General point throughout – the brightness and number of Sec7 foci varies a lot (not specific to tag or microscope), which is also true in the literature. As whether compartments are either Sec7 + or – is so important for this study can this variability be quantified or discussed in the text. Does it vary based on the cell cycle, age of cell, media conditions, growth phase, etc?

Figure 1

The variation in localization of Tlg2 (even using the same 2D imaging technique) is disconcerting. GFP-Tlg2 localises exclusively to discrete foci in Figure 1A but mainly seems to be dispersed in a large diffuse region in Figure 2A. This difference is obvious when only comparing two representative images – how common is that diffuse signal and where in the cell does it represent (maybe intravacuolar? This could be tested with dyes like FM4-64 or CMAC)? These are never observed in the SCLIM micrographs, possibly the contrast from processing enhances bright signal at the cost of molecules from low/diffuse regions?

Line 145 – "localized at puncta around the vacuole and vacuole" confused me initially, it might read better as "localized inside the vacuole and at puncta around the vacuole"

Figure 2A – The inset showing 16-24 seconds. Why is Abp1 label pink in A but red in B – this should probably be consistent.

Figure 2I – Many Abp1+ endocytic carries are very large/bright whilst some are very small. Are they all vesicles or are some indictors of fusion with non-TGN compartments? Also, even without being colour-blind I struggle to distinguish the Sec7-iRFP and GFP-Tlg2 signal, it might be improved to show a third micrograph with only one of them?

Line 218 – The endocytosis of vesicles to Tlg2+ compartment is very convincing. It seems important for the segue from Figure 1 to also image internalised mating factor in cells with labelled Abp1. I mention this because further clarification about what endocytic carriers do/do not have cargo seems to be an outstanding issue that can be reconciled with this data (see above point relating to figure 2I).

Line 255 – what is meant by the "early-to-late stage of the endocytic pathway"? There are imaging experiments with predefined criteria, such as the differences in (co)localisation of A647 mating factor at 5 and 15 minutes, maybe these would be more specific reference points?

Figure 3E – I think the bottom row of micrographs in 3E in mislabelled – it makes more sense for it to be labelled V (in red) and S (cyan)? Instead of T (green) and S (cyan).

Figure 3J – The bottom left cell in has an obvious 5 min internalised a-factor foci that is distinctly Tlg2-negative. How common is this observation in raw data? Is this a distinct endosomal compartment? If ideas like this based on the representative micrographs are valid, they could be discussed.

Line 312 – I think the rationale for these experiments is good but not explained sufficiently for non-experts. I think more information and references should be included about Golgi clathrin adaptors (eg Duncan et al. 2003 Nat. Cell Bio.) This will better contextualise the results relating to Gga/Ent/AP mutants and their impact on endocytic cargo trafficking.

Figure 5A – As mentioned in main summary, there are inconsistent localisations of these important reporter proteins that are not mentioned or explained. GFP-Tlg2 is clearly on the vacuolar membrane in 5A but not 5E. Are the cells grown in different conditions or experiencing stress (e.g. nutrient stress) in certain imaging experiments? A brightfield image could be included to demonstrate that no gross aberrations have occurred prior to imaging.

Figure 6A – I was a little confused as to whether all the pheromone+ and Snc1+ compartments are all accounted for across time lapse imaging. The TIRF microscopy appears to show both (Tlg2+ pheromone-) and (Tlg- pheromone+) foci. Are these distinct carriers and what drives the cargo partitioning?

Figure 7 – Although the markers Tlg2 and Snc1 are convenient to label structures being studied, they also represent core machinery and cognate SNARE partners for these trafficking pathways. This could be expounded in the text, both in the results and discussion. Related, does the imaging in Figure 6 fully support the model shown in Figure 7 relating to the SNARE cycle and when Snc1-Tlg2 are, and are not, colocalised?

Line 435 – Obviously the Day et al. model needs to be discussed here and the inclusion of earlier work regarding Tlg1 (especially that of Howard Riezman) is helpful. This section could also be an opportunity to discuss the yeast endosomal field in more detail. For example, lattice light sheet experiments showing fast-moving peripheral Vps4+ endosome populations are distinct from the TGN and PVC which have been proposed as early/sorting endosomes (PMIDs: 29019322 + 36125415). Could these intermediates be the same as the Tlg2+ compartments uncovered in this study?

Line 474 – From a semantic point, if endocytosed cargo enters a Tlg2+ compartment en route to later endosomes marked with Vps21 can it be considered an endosome as much a TGN sub-compartment?

---

## [Author Response]

Essential revisions:Reviewer #1 (Recommendations for the authors):The article is very well written and the findings are well presented. I have the following suggestions:1. Because the yeast TGN has been suggested to function not only as an early endosome, but also as the recycling endosome, the study feels incomplete without an analysis and clear discussion of the recycling endosome. The authors should try to place the recycling endosome as well in their model of how the TGN relates to endocytosis by conducting a similar analysis of Rab11 or other markers in relation to Sec7, Tlg2, Snc1 and endocytosed/recycled cargo.

In accordance with the reviewer's suggestion, we have examined the dynamics of Ypt31p and Ypt32p (yeast Rab11 homologues) with at the Tlg2p- or Sec7p-residing compartment. We first confirmed that Ypt31p and Ypt32p exhibit almost the same localization and dynamics (Figure 7—figure supplement 1, B-D), consistent with previous studies (Gingras et al., *eLife*, 2022). We next compared the localization of Alexa594-labeled a-factor (A594-a-factor) with GFP-Ypt31p and found that Ypt31p does not highly overlap with A594-a-factor (Figure 7—figure supplement 1, E and F). We also performed triple-color 4D imaging to determine whether Ypt31p localized to the Tlg2p or the Sec7p sub-compartment. As reported previously, in the wild-type cell Ypt31p appears at the TGN after Sec7p (Suda et al., PNAS. 2013; Highland and Fromme, MBoC, 2021). However, the signal partially overlaps with both the Tlg2p and Sec7p signals (Figure 7, A-D). Line scan analysis of the TGN, which includes Tlg2p, Sec7p and Ypt31p, revealed that the mCherry-Ypt31p signal partially overlaps with both the Sec7-iRFP and GFP-Tlg2p signals (Figure 7C). Additionally, 4D SCLIM imaging demonstrated that Ypt31p appears at, and disappears from the TGN, with timing similar to that of Snc1p, indicating that Ypt31p resides in the recycling (or sorting) compartment (Figure 7, E-H). These results suggest that Ypt31p resides in both the Tlg2p and Sec7p compartments, and presumably functions in the secretory and recycling pathways. In contrast, in the *gga1*D *gga2*D mutant, GFP-Ypt31p signals were little overlapped with the mCherry-Snc1p signals (Fig, 7, I-L). Since deletion of Gga1p/2p affects turnover of the Tlg2p sub-compartment (Figure 5, F and G), the recruitment of Ypt31p seems to fail in the mutant. Based on these observations, in the new manuscript we have added Ypt31p localization to the model (Figure 8) and also provided some discussion of these findings (lines 556-569).

2. I would suggest not to use the cyan/green color combination to make the point that markers overlap, as this could lead to visual overestimation of that overlap (e.g. Figure 1F, 2J, 3G, 4J, 5K, 6C).

According to reviewer’s suggestion, we have changed color from cyan to blue in Figure 1E-H, 2I, J, 3D-G, H, 4I, J, 5J-L, 6B-I in the new manuscript. We have also changed color in Video 1, 2, 6, 7, 11, 12, and 13.

3. Line 57: "Additionally, we have previously demonstrated that endocytosed cargos are rarely transported to the Sec7p-residing TGN compartment (Toshima et al., 2014)." This needs some better description and context, as the present study shows that endocytosed cargo travels to the Tlg2 sub-compartment, which overlaps with the Sec7 sub-compartment. Should the authors talk about "Tlg2-only" and "Sec7-only" sub-compartments? However, see next point.

According to the reviewer's suggestion, we have changed the sentence “~ endocytosed cargos are rarely transported to the Sec7p-residing TGN compartment” to “~ endocytosed cargos are only slightly co-localized with Sec7p-containing TGN cisternae classified into the early-to-late TGN” in the new manuscript (lines 59-61). We have also cited a study by Tojima et al. to explain that the TGN is classified into the early and late TGN.

4. Line 140: "In contrast, Sec7-GFP, which was also detected as several punctate structures, did not show significant overlap with A594-α-factor signals either at 5 min (26.0{plus minus}9.4%) or 10 min (22.7{plus minus}6.0%) after A594-α-factor internalization (Figure 1B; Figure S1A)." In what sense are 26.0 % and 22.7 % overlap between Sec7 and α-factor "not significant"?

As the reviewer pointed out, we agree that the word “not significant” in this sentence is not appropriate. In the new manuscript, we have changed this sentence to "In contrast, Sec7-GFP, which was also detected as several punctate structures, showed considerably lower overlap with A594-α-factor signals than Tlg2-GFP ~” (lines 145-146).

5. In the Discussion section called "Integrating the present model with earlier observations", the authors should discuss the findings on Snc1/Tlg1 in Best et al., 2020, and how they can be integrated with those of Day et al., 2018 and their own.

We appreciate the reviewer’s helpful suggestion. According to the reviewer’s suggestion, we have cited the study by Best et al., and discussed how their studies can be integrated to our findings (lines 618-628).

6. There is no attempt to put the findings in an evolutionary context. With the exception of some references to their own work in plants, there is no discussion of the implications for other organisms, such as *Drosophila* or humans.

According to the reviewer's suggestion, we have added a paragraph (Endocytic recycling compartments in different organisms) discussing our finding in an evolutionary context in the new manuscript (lines 630-649).

Reviewer #2 (Recommendations for the authors):General point throughout – the brightness and number of Sec7 foci varies a lot (not specific to tag or microscope), which is also true in the literature. As whether compartments are either Sec7 + or – is so important for this study can this variability be quantified or discussed in the text. Does it vary based on the cell cycle, age of cell, media conditions, growth phase, etc?

In accordance with the reviewer’s suggestion, we examined the number of Sec7p-residing puncta at different stages of the cell cycle and different culture media. All the data were obtained using cells in the logarithmic growth phase (OD600 = ~0.5). Two different media, YPD and SD (with amino acids), were used in this study, but no obvious difference in the number of Sec7p-residing puncta was observed under the growth conditions employed (data not shown). In contrast, the number of Sec7p-residing puncta were slightly more abundant in M phase cells than in S phase cells (Figure 5—figure supplement 1, A and B). However, the rate of Sec7-mCherry overlapping with GFP-Tlg2p was almost the same between S phase and M phase cells (Figure 5—figure supplement 1, C and D), suggesting that our present results were not affected by the cell cycle or media conditions. To clarify this, we have added some explanatory sentences on lines 379-385 in the new manuscript.

Figure 1The variation in localization of Tlg2 (even using the same 2D imaging technique) is disconcerting. GFP-Tlg2 localises exclusively to discrete foci in Figure 1A but mainly seems to be dispersed in a large diffuse region in Figure 2A. This difference is obvious when only comparing two representative images – how common is that diffuse signal and where in the cell does it represent (maybe intravacuolar? This could be tested with dyes like FM4-64 or CMAC)? These are never observed in the SCLIM micrographs, possibly the contrast from processing enhances bright signal at the cost of molecules from low/diffuse regions?

We agree that GFP-Tlg2p exhibits diffused localization in the cytosol, in addition to punctate localization. In accordance with the reviewer’s suggestion, we labeled cells with FM4-64 and found that the diffuse localization occurred in the vacuolar lumen (Figure 1—figure supplement 1A). As previous studies have reported that Tlg2p is cycled between the TGN and the endosome (Lewis et al., MBoC, 2000), we speculate that Tlg2p is partially transported to the vacuole. This vacuolar localization was not visible in some cells, possibly due to the location of the vacuole. Thus, we have added explanatory sentence about this in the new manuscript (lines 139-143). As the reviewer mentions, this diffuse localization in the vacuolar lumen was not evident by SCLIM presumably because the fluorescence signal was too weak. To explain these points, we have added some sentences on lines 180-182 in the new manuscript.

Line 145 – "localized at puncta around the vacuole and vacuole" confused me initially, it might read better as "localized inside the vacuole and at puncta around the vacuole"

According to the reviewer's suggestion, we have changed the sentence “localized at puncta around the vacuole and vacuole” to “localized inside the vacuole and at puncta around the vacuole” in the new manuscript (lines 151-152).

Figure 2A – The inset showing 16-24 seconds. Why is Abp1 label pink in A but red in B – this should probably be consistent.

We appreciate the reviewer’s pointing out our error, which we have corrected.

Figure 2I – Many Abp1+ endocytic carries are very large/bright whilst some are very small. Are they all vesicles or are some indictors of fusion with non-TGN compartments?

Since Abp1-mCherry labels F-actin associated with endocytic vesicles, the fluorescence intensity depends on the amount of actin polymerization. Additionally, some Abp1-mCherry signals fuse with each other and become large. To explain this, in the new manuscript, we have added the data showing that the Abp1-mCherry signal changes over time and that some Abp1-mCherry patches fuse with each other (Figure 2—figure supplement 1, B-D).

With regard to the question "are some indictors of fusion with non-TGN compartments?", since the rate of actin depolymerization is not constant for each endocytic vesicle, it is likely that some Abp1-mCherry signals disappear before reaching the endosome. Therefore, it was difficult to show that all vesicles fuse with Tlg2p-residing compartments. We have added an explanatory sentence about this possibility in the new manuscript (line 233-235).

Also, even without being colour-blind I struggle to distinguish the Sec7-iRFP and GFP-Tlg2 signal, it might be improved to show a third micrograph with only one of them?

According to the reviewer's suggestion, we have added single color micrograph to show Sec7-iRFP localization. Additionally, we have changed color of Sec7-iRFP from cyan to blue to clearly separate Sec7-iRFP and GFP-Tlg2 signals (Figure 2, I and J).

Line 218 – The endocytosis of vesicles to Tlg2+ compartment is very convincing. It seems important for the segue from Figure 1 to also image internalised mating factor in cells with labelled Abp1. I mention this because further clarification about what endocytic carriers do/do not have cargo seems to be an outstanding issue that can be reconciled with this data (see above point relating to figure 2I).

In our previous study, using TIRF microscopy, we have demonstrated that Alexa Fluor 488-labeled a-factor accumulates at the Abp1p patch at the cell surface, and is internalized concomitantly (Toshima et al., PNAS, 2006). However, we have not yet succeeded in capturing an image showing an internalized Abp1p-labeled endocytic vesicle carrying fluorescence-labeled a-factor, presumably because the a-factor fluorescent signal is below the limit of detection. Therefore, we have cited the study and added an explanatory sentence to suggest that the Abp1p-labeled vesicles contain cargos (lines 223-225).

Line 255 – what is meant by the "early-to-late stage of the endocytic pathway"? There are imaging experiments with predefined criteria, such as the differences in (co)localisation of A647 mating factor at 5 and 15 minutes, maybe these would be more specific reference points?

We appreciate the reviewer’s helpful suggestion. We have changed the sentence “at the early-to-late stage of the endocytic pathway” to “at 5-10 min after internalization (Figure 1B)” in the new manuscript (line 280).

Figure 3E – I think the bottom row of micrographs in 3E in mislabelled – it makes more sense for it to be labelled V (in red) and S (cyan)? Instead of T (green) and S (cyan).

We appreciate the reviewer’s pointing out our errors, which we have corrected.

Figure 3J – The bottom left cell in has an obvious 5 min internalised a-factor foci that is distinctly Tlg2-negative. How common is this observation in raw data? Is this a distinct endosomal compartment? If ideas like this based on the representative micrographs are valid, they could be discussed.

In our previous study, we demonstrated that in the vps21D ypt52D mutant A594-a-factor accumulates at the endosomal intermediates labeled by Vph1-GFP (Toshima et al., Nat Comm, 2014). In the present study, we showed that Tlg2p also accumulated in the endosomal intermediates and that the number of GFP-Tlg2p-labeled puncta increased in the vps21D ypt52D mutant (Figure 3J). As the reviewer points out, A594-a-factor labeled Tlg2-negative puncta are frequently observed in the vps21D ypt52D mutant (Figure 3J), and we speculate that these puncta are probably endosomal intermediates that do not contain GFP-Tlg2p. Thus, in accordance with the reviewer’s suggestion, we have added the following sentence: "We also observed that A594-a-factor accumulates at endosomal intermediate-like structures that do not contain the GFP-Tlg2p signal in the vps21D ypt52D mutant (Figure 3J, white arrowhead)." in lines 320-323.

Line 312 – I think the rationale for these experiments is good but not explained sufficiently for non-experts. I think more information and references should be included about Golgi clathrin adaptors (eg Duncan et al. 2003 Nat. Cell Bio.) This will better contextualise the results relating to Gga/Ent/AP mutants and their impact on endocytic cargo trafficking.

We appreciate the reviewer’s helpful suggestion. According to the reviewer's suggestion, we have cited the studies that the reviewer advised and added the sentences to explain about Golgi clathrin adaptors in lines 333-341 in the new manuscript.

Figure 5A – As mentioned in main summary, there are inconsistent localisations of these important reporter proteins that are not mentioned or explained. GFP-Tlg2 is clearly on the vacuolar membrane in 5A but not 5E. Are the cells grown in different conditions or experiencing stress (e.g. nutrient stress) in certain imaging experiments? A brightfield image could be included to demonstrate that no gross aberrations have occurred prior to imaging.

As the reviewer points out, GFP-Tlg2p is slightly localized at the vacuolar membrane in the *gga*1D *gga*2D mutant (Figure 5A), suggesting that retrograde transport of Tlg2p from the endosome to the TGN is partially impaired in this mutant. In the new manuscript, we have added a brightfield (DIC) image (Figure 5A) and mentioned about this (lines 389-391). The vacuolar membrane localization was not observed by SCLIM because the fluorescent signal was too weak to be detected. Therefore, to explain this, we have added a sentence to the new manuscript (lines 399-400).

Figure 6A – I was a little confused as to whether all the pheromone+ and Snc1+ compartments are all accounted for across time lapse imaging. The TIRF microscopy appears to show both (Tlg2+ pheromone-) and (Tlg- pheromone+) foci. Are these distinct carriers and what drives the cargo partitioning?

As the reviewer points out, TIRFM imaging showed that three types of foci (A594-a-factor+/GFP-Snc1p+, A594-a-factor+/GFP-Snc1p-, and A594-a-factor-/GFP- Snc1p+) exist on the cell surface. In the previous study, we demonstrated that A594-a-factor only partially colocalized with GFP-Snc1p when it was endocytosed (Toshima et al., PNAS, 2006). One possible explanation for why not all GFP-Snc1-labeled vesicles include the A594-a-factor signal (A594-a-factor-/ GFP-Snc1p+ foci) is that some vesicles do not carry enough A594-a-factor to allow detection of the fluorescence signals. As for (A594-a-factor+/GFP-Snc1p-) foci, GFP-Snc1p may not be loaded on all endocytic vesicles because GFP-Snc1p is exogenously expressed and endogenous Snc1p exists in the cell. We agree that what drives the cargo partitioning is quite an important issue, but we were unable to address it in this study. Therefore, we have added a sentence to explain why several kinds of foci containing distinct cargo were observed (lines 441-444).

Figure 7 – Although the markers Tlg2 and Snc1 are convenient to label structures being studied, they also represent core machinery and cognate SNARE partners for these trafficking pathways. This could be expounded in the text, both in the results and discussion. Related, does the imaging in Figure 6 fully support the model shown in Figure 7 relating to the SNARE cycle and when Snc1-Tlg2 are, and are not, colocalised?

According to the reviewer's suggestion, we have added the explanatory sentence “~ in the present study that the compartment where Snc1p’s cognate Q-SNARE, Tlg2p, resides, is the first destination for endocytic traffic, ~" in results (lines 431-432). We have also added the sentence "We have also showed that Snc1p, endocytic R-SNARE that targets Tlg2p ~" in discussion (lines 542-543).

In Figure 6D, we showed that Tlg2p appears at the TGN earlier than Snc1p and disappears from there before Snc1p. This suggests that Snc1p binds to pre-existing Tlg2p, which is recycled back from the PVC, and dissociates from Tlg2p at the Tlg2p sub-compartment. This result is also supported by 3D triple-color observation using mCherry-Snc1p, GFP-Ypt31p and Sec7-iRFP (Figure 7F). To explain about the SNARE cycle in Figure 8 (in the new manuscript), we have added sentences "As shown in Figure 6, Tlg2p appears at TGN earlier than Snc1p and disappears before Snc1p from the TGN. This suggests that Snc1p binds to Tlg2p recycled back from the PVC and dissociate from Tlg2p at the Tlg2p sub-compartment." in the discussion (lines 545-548).

Line 435 – Obviously the Day et al. model needs to be discussed here and the inclusion of earlier work regarding Tlg1 (especially that of Howard Riezman) is helpful. This section could also be an opportunity to discuss the yeast endosomal field in more detail. For example, lattice light sheet experiments showing fast-moving peripheral Vps4+ endosome populations are distinct from the TGN and PVC which have been proposed as early/sorting endosomes (PMIDs: 29019322 + 36125415). Could these intermediates be the same as the Tlg2+ compartments uncovered in this study?

We appreciate the reviewer kindly providing information regarding papers that we should cite. In accordance with the reviewer’s suggestion, we have added some discussion about the study by Day et al. (Dev Cell, 2018) and by Dr, Riezman (Prescianotto-Baschong and Reizman, MBoC, 1998) in lines 533-542. We have also discussed about the Vps4+ endosome by citing the studies (Laidlaw et al., JCB, 2022; Adell et al., *eLife*, 2017) in that section (lines 550-555). We have further added a new Discussion section about early and recycling endosomes in yeast and other organisms in lines 621-640 (please see also the response to reviewer #1’s comment #5). As Vps4-positive endosomes are known to be distinct from Snc1p-positive structure in the study by Laidlaw et al., it is likely that these endosomes are not same as the Tlg2-residing compartment. In the new manuscript, we have also mentioned about this (lines 552-555).

Line 474 – From a semantic point, if endocytosed cargo enters a Tlg2+ compartment en route to later endosomes marked with Vps21 can it be considered an endosome as much a TGN sub-compartment?

Since we think that it is not appropriate to consider the Tlg2-residing compartment adjacent to the Sec7-residing compartment as endosome, we have referred to this compartment as "early/sorting compartment" throughout the manuscript.